

# The OCO-3 mission; measurement objectives and expected performance based on one year of simulated data

Annmarie Eldering[1], Tommy E. Taylor[2], Chris W. O'Dell[2], and Ryan Pavlick[1]

[1]Jet Propulsion Laboratory, California Institute of Technology, Pasadena, CA, 91109, USA.
[2]Cooperative Institute for Research in the Atmosphere, Colorado State University, Fort Collins, CO, 80521, USA.

**Correspondence:** Annmarie Eldering (eldering@jpl.nasa.gov)

**Abstract.** The Orbiting Carbon Observatory-3 (OCO-3) is NASA's next instrument dedicated to extending the record of the dry-air mole fraction of column carbon dioxide ($XCO_2$) and solar-induced fluorescence (SIF) measurements from space. The current schedule calls for a launch in the first half of 2019 via a Space-X Falcon 9 and Dragon capsule, with installation as an external payload on the Japanese Experimental Module Exposed Facility (JEM-EF) of the International Space Station (ISS). The nominal mission lifetime is 3 years. The precessing orbit of the ISS will allow for viewing of the earth at all latitudes less than approximately $52°$, with a ground repeat cycle that is much more complicated than the polar orbiting satellites that so far have carried all of the instruments capable of measuring carbon dioxide from space.

The grating spectrometer at the core of OCO-3 is a direct copy of the OCO-2 spectrometer, which was launched into a polar orbit in July 2014. As such, OCO-3 is expected to have similar instrument sensitivity and performance characteristics to OCO-2, which provides measurements of $XCO_2$ with precision better than 1 ppm at 3 Hz with each viewing frame containing 8 footprints of approximate size 1.6 by 2.2 km. However, the physical configuration of the instrument aboard the ISS, as well as the use of a new pointing mirror assembly (PMA), will alter some of the characteristics of the OCO-3 data, compared to OCO-2. Specifically, there will be significant differences from day to day in the sampling locations and time of day. In addition, the flexible PMA system allows for a much more dynamic observation mode schedule.

This paper outlines the science objectives of the OCO-3 mission and, using a simulation of one year of global observations, characterizes the spatial sampling, time of day coverage, and anticipated data quality of the simulated L1b. After application of cloud and aerosol prescreening, the L1b radiances are run through the operational L2 full physics retrieval algorithm, as well as post-retrieval filtering and bias correction, to examine the expected coverage and quality of the retrieved $XCO_2$ and to show how the measurement objectives are met. In addition, results of the SIF from the IMAP-DOAS algorithm are analyzed. This paper focuses only on the nominal nadir-land and glint-water observation modes, although on-orbit measurements will also be made in transition and target modes, similar to OCO-2, as well as the new "snapshot" area mapping mode.

## 1 Introduction

As called for in NASA's Climate Architecture Report (June 2010), the Orbiting Carbon Observatory-3 (OCO-3) was built from spare parts during the construction of OCO-2 to be made available as an instrument of opportunity. After assessment of various



options, the decision was made in 2013 to design and build the OCO-3 payload for operation on the International Space Station (ISS). The primary scientific objective of OCO-3 is to provide global, dense, high-precision measurements of the dry-air mole fraction of column carbon dioxide ($XCO_2$) and solar-induced fluorescence (SIF) from space. A planned 3 year lifetime aboard the ISS will allow for continuation of the international measurement record of $CO_2$ that began in earnest with the Japanese

GOSAT satellite (January 2009 to present) (Kuze et al., 2009), followed by the NASA OCO-2 mission (July 2014 to present), and most recently by the Chinese TANSAT (December 2016 to present) (Yang et al., 2018). Furthermore, a 2019 launch of OCO-3 would provide overlap for future planned missions such as GOSAT-2 (planned 2018 launch) (Nakajima et al., 2012), the MicroCARB mission from CNES (planned 2021 launch) (Buil et al., 2011), and possibly even with the recently selected NASA GeoCarb mission (O'Brien et al., 2016), which has a planned 2022 launch. Because of the relatively small variations

in atmospheric $CO_2$ globally, it is critical to understand how the data products from various sensors intercompare at levels less than their precision, which is 0.1% for both OCO-2 and OCO-3. It is worth noting that all of the sensors mentioned above are polar orbiting, with the exception of OCO-3 (precessing) and GeoCarb, which is the first planned geostationary observation system for measuring $XCO_2$.

The nominal planned viewing strategy of OCO-3 is to take down-looking nadir viewing measurements over land to minimize

the probability of cloud and aerosol contamination. Over water measurements will be taken near the specular reflection spot (glint viewing) to maximize the signal over the low reflectivity surface. However, unlike OCO-2, which performs complex maneuvers of the entire satellite bus to observe ground targets, the OCO-3 instrument will be fitted with an agile 2-D pointing mechanism, i.e., a pointing mirror assembly (PMA). This will allow for rapid transitions between nadir and glint mode (less than 1 minute). The PMA will also allow for target mode observations, similar to those taken by OCO-2, typically at Total

Column Carbon Observation Network (TCCON) ground sites for use in validation (Wunch et al., 2010). The PMA will provide the ability to scan large contiguous areas (order 80 km by 80 km), such as cities and forests, on a single overpass. This will be known as "snapshot" mode and will allow for fine scale spatial sampling of $CO_2$ and SIF variations unlike what can be done with any current satellite system. If OCO-2 and OCO-3 operate concurrently, the snapshot mode can be used to gather a significant fraction of overlapping data. However, this paper deals exclusively with the two main viewing modes (nadir-land

and glint-water), while a detailed discussion of snapshot mode is deferred to a companion paper.

The sampling that will be provided by OCO-3 aboard the precessing ISS will differ significantly compared to the polar orbits of OCO-2 and GOSAT. The overpasses will not always occur at the same local time of day for a given point on the earth, which has implications with respect to the diurnal cycle of both clouds and aerosols (which contaminate the observations of $XCO_2$) and studies of the carbon cycle, which itself has a strong diurnal variation. The precession in time-of-day sampling

will be especially informative for the SIF observations with respect to studying the biosphere response (both natural and anthropogenic) to changes in sunlight.

The international record of satellite remote sensing of $CO_2$ has extended across a number of measurement platforms, e.g. SCIAMACHY (2002-2012), Aqua-AIRS (2002-present), GOSAT (2009-present), TANSAT (2016-present) and is being used to quantify several aspects of the carbon cycle. The $CO_2$ seasonal cycle has been studied with SCIAMACHY and GOSAT

data (e.g., (Buchwitz et al., 2015; Lindqvist et al., 2015; Reuter et al., 2013; Wunch et al., 2013)). The GOSAT measurements





have been used to characterize a number of relatively large disturbances to the carbon cycle, including reduced carbon uptake in 2010 due to the Eurasia heat wave (Guerlet et al., 2013), larger than average carbon fluxes in tropical Asia in 2010 due to above-average temperatures (Basu et al., 2014), and anomalous carbon uptake in Australia (Detmers et al., 2015). In addition, Parazoo et al. used GOSAT $XCO_2$ and SIF estimates to better understand the carbon balance of southern Amazonia (Parazoo

et al., 2014), while Ross et al. used GOSAT data to obtain information on wildfire $CH_4$:$CO_2$ emission ratios (Ross et al., 2013).

Relative to earlier carbon dioxide measurements from space, OCO-2 is providing a much denser data set (in both time and space) with higher precision in retrieved $XCO_2$. The publicly available B7 version of the OCO-2 data ((https://disc.gsfc.nasa.gov/)) has recently been used to quantify changes in tropical carbon fluxes (Liu et al., 2017) and the equatorial Pacific ocean (Chatterjee et al., 2017), due to the strong 2015 El Nino. Schwandner et al. (2017) highlighted localized sources detected by OCO-2,

while Eldering et al. (2017b) provided an extensive global view of the atmospheric carbon dioxide as observed from OCO-2 after its first two years in space.

In order to continue the international measurement record of global carbon dioxide from space, NASA plans to operate OCO-3 from the ISS for a period of about 3 years, beginning nominally in early 2019. Since there are a number of new considerations related to the unique viewing and sampling from this platform, it is desirable to study the expected performance

of the instrument prior to launch. To do this we generated one full year of simulated OCO-3 measurements, on which we ran the current versions of the OCO-2 prescreeners and L2 retrieval, as well as the post-processing quality filtering and bias correction. The bulk of this paper is based on these simulations to evaluate expected data quality and data density from OCO-3 aboard the ISS.

The paper is organized as follows. Section 2 provides an overview of the OCO-3 mission, the science objectives and planned

measurement modes. In Section 3, the generation of one year of simulated L1b radiances using realistic geometry, instrument characteristics and meteorology is detailed. Section 4 discusses properties of the retrieved $XCO_2$ from OCO-3 using effectively operational algorithms for retrieval, filtering, and bias correction. Overall analysis of the results are presented in Section 5. Particular focus is given to the temporal and spatial coverage, expected signal-to-noise ratios, and $XCO_2$ and SIF errors. Finally, Section 6 provides a summary of the expected performance of the OCO-3 mission based on these simulations.

## 25 2 The OCO-3 science objectives and measurement overview

Like OCO-2, the OCO-3 mission has been designed to collect a dense set of precise measurements of $XCO_2$ with a small footprint. The scientific objective of the mission is to quantify variations of $XCO_2$ with the precision, resolution, coverage, and temporal stability needed to improve our understanding of surface sources and sinks of carbon dioxide on regional scales ($\simeq 1000\,\mathrm{km}$ by $1000\,\mathrm{km}$) and the processes controlling their variability over the seasonal cycle. The measurement objective

is to quantify the dry air column carbon dioxide ratio (the total column of carbon dioxide normalized by the column of dry air) to better than 1 ppm for collections of 100 footprints, the same objective as OCO-2. The footprint size is equal to or less then $4\,\mathrm{km}^2$, and changes in aspect ratio with the viewing geometry. The OCO-3 mission will also provide a measurement of solar-induced fluorescence (SIF), again with similar characteristics as OCO-2. As will be discussed in Section 3, the sampling



characteristic from the ISS will result in changing latitudinal coverage each month, such that the regions where sources and sinks can be quantified will vary in time. The nominal measurement operation mode will be to collect data in nadir viewing over land and glint viewing over oceans, with a variable number of target and snapshot mode measurements integrated each day.

In addition, the OCO-3 mission also has the potential to contribute to carbon cycle science beyond its primary objective. The current plan includes the nearly simultaneous installation of three other instruments aboard the ISS that are focused on various aspects of the terrestrial carbon cycle (Stavros et al., 2017). This includes NASA's Global Ecosystem Dynamics Investigation (GEDI), which is a lidar instrument designed to make observations of forest vertical structure to assess the above-ground carbon balance of the land surface and investigate its role in mitigating atmospheric $CO_2$ in the coming decades (Dubayah et al., 2014;

Stysley et al., 2015). NASA/JPL's Ecosystem Spaceborne Thermal Radiometer Experiment on Space Station (ECOSTRESS) will measure evapotranspiration and assess plant stress and its relationship to water availability (Fisher et al., 2015; Hulley et al., 2017) Finally, the Hyperspectral Imager Suite (HISUI) from JAXA will have a multi-band spectrometer with a focus on identifying plant types (Matsunaga et al., 2015, 2017). The integration of these data, along with OCO-3 measurements of $XCO_2$ and SIF, have the potential to inform our understanding of many aspects of ecosystem processes (Stavros et al., 2017).

An additional enhancement to the OCO-3 data set will be provided by the currently operating OCO-2 instrument if its special pointing capability is synchronized with this suite of instruments to view specific ground targets. A second opportunity for OCO-3 relates to the use of the snapshot mode to focus on emissions hotspots, such as emissions from cities and power plants, or from natural sources such as volcanoes and wildfires. If OCO-2 and OCO-3 operate concurrently, complementary sampling could maximize the insights on the sources and sinks of carbon dioxide.

## 2.1    The OCO-3 instrument payload

At the core of OCO-3 is a three-band grating spectrometer built as a spare for the OCO-2 instrument, which measures reflected sunlight (Crisp et al., 2017; Eldering et al., 2017a). The oxygen A-band ($O_2$ A-band) measures absorption by molecular oxygen near $0.76\,\mu$m, while two carbon dioxide bands, labelled here as the weak and strong $CO_2$ bands, are located near 1.6 and $2.0\,\mu$m, respectively. The $O_2$ A-band is sensitive to the atmospheric path length observed due to the absorption by oxygen,

and is used to estimate the apparent surface elevation and as part of the detection of clouds. This band also provides sensitivity to solar-induced fluorescence (SIF), a small amount of light emitted during plant photosynthesis (Frankenberg et al., 2012). The weak $CO_2$ and strong $CO_2$ bands provide sensitivity to carbon dioxide, with peaks at different vertical heights, but are also used as part of the cloud detection scheme since they are particularly sensitive to the wavelength dependence of aerosol scattering and absorption. Estimates of $XCO_2$ are derived from these spectra using an optimal estimation retrieval method that

integrates detailed models of the physics of the atmosphere (Bösch et al., 2006; Connor et al., 2008; O'Dell et al., 2012).

The instrument has 1016 spectral elements in each band, with 160 pixels averaged in groups of 20 along the slit, creating eight spatial footprints. The entrance optics have been modified to reduce the magnification from 2.4:1 to 1:1, to maintain similar footprint sizes given the lower altitude of the ISS, which typically flies at $\simeq 404\,$km, compared to OCO-2 at $\simeq 705\,$km. This magnification change will result in OCO-3 footprints that are $< 4\,\text{km}^2$, comparable to the $3\,\text{km}^2$ of OCO-2. The instrument





field of view, i.e., the frame, will be approximately 13 km, or 1.6 km width per eight footprints, and the spacecraft motion covers ≃2.2 km during the 0.33 seconds of integration time. The rate of data collection will be approximately 1 million sets of 3 spectral band measurements per day, before considering ISS limitations discussed in Section 2.3.

The OCO-3 project inherited a fully characterized spectrometer from the OCO-2 project, which was designed for integration on a LeoStar spacecraft. For utilization on the ISS JEM-EF, a number of adaptations were required (Basilio et al., 2013). These include redesign of the thermal system, updates to the electrical system, and updates to the data flow from the instrument to the data processing center at JPL. These changes do not fundamentally change the radiometric characteristics, and therefore the science data quality, so will not be discussed in this paper. As described in the following section, a new pointing mirror assembly was also required for OCO-3.

## 2.2 OCO-3 pointing mirror assembly overview

A design change that impacts the radiometric characteristics of the data is the addition of a pointing mirror assembly (PMA). The PMA is required to allow non-nadir observations from the fixed position on the ISS, unlike the currently operating OCO-2, which maneuvers the entire spacecraft to point. Two important design requirements of the PMA were to allow quick movement through a large range of angles, and that the movement not impart any angular dependent polarization or radiance changes in the measurements. To meet these objectives a variation of the pointing system designed for the Glory Aerosol Polarimetry Sensor (APS) (Persh et al., 2010) was selected. This system relies on a single pair of matched mirrors in an orthogonal configuration that impart less than 0.05% change to the polarization (Mishchenko et al., 2007). For the OCO-3 PMA the concept was extended to a 2-axis pointing system. There are two elements; one controlling the azimuthal (cross-track) angle, and the other controlling the elevation (along-track) angle. Although the PMA itself does not change the polarization of the light more them 0.1%, there are polarization implications, since the image of the slit is rotated as a function of the change in the PMA, primarily driven by the elevation (along-track) angle. It is worth noting that reflected sunlight is naturally polarized by its interaction with the earth's surface and atmosphere, especially over water.

Early in the mission design, a trade study was performed, evaluating the expected signal with and without the installation of an additional polarization scrambler, i.e., a polarization nulling optical component. Without a scrambler, the signal measured by the instrument is large when the polarization of the incident light is aligned with the axis of polarization sensitivity of the instrument, i.e, polarization angle of zero. However, the measured signal decreases towards zero as the polarization of the incoming light rotates orthogonal to the axis of instrument polarization (polarization angle of 90 degrees). Inserting a scrambler would make the polarization orientation of the incoming light random, regardless of the position of the PMA, but would reduce the signal by nearly 50% compared to having no scrambler. In addition, the analysis showed that a single optical element that could scramble light at all of the OCO-3 wavelengths could not be manufactured and characterized to the required precision. Lastly, the volume and coverage of data with sufficient signal in the no-scrambler case was predicted to be more than sufficient to meet the science objectives. Therefore, OCO-3 will be operated without a polarization scrambler.

As will be discussed in more detail in Section 5.1, the PMA is one element that contributes to a change in the overall light throughput of OCO-3 compared to OCO-2. In the $O_2$ A-band, each mirror has a reflectivity of 95.4%, so the 4 mirror PMA





system has an effective transmission of 83%. The weak and strong $CO_2$ band overall transmissions are higher, at 93% and 95%, respectively.

The thermal vacuum testing of OCO-3 has been finalized and analysis of the data is underway, with completion planned in the second half of 2018. Details of the measurement performance and how it compares to the instrument requirements will be reported in a forthcoming manuscript.

## 2.3 Sampling from the International Space Station - routine measurements

The ISS orbit is nearly circular about the earth, with altitudes that ranges from 330 km to 410 km. The planned altitude during the time of OCO-3 operation is 405 km. With a ground-track velocity of 27,600 km/h (7.667 km/s), one orbit around the earth is completed in about 92 minutes. The inclination of the orbit is 51.6°, which limits the latitudinal range that can be sampled by OCO-3. These orbital parameters result in a precessing orbit, meaning the local overpass time varies across all hours of the day over the course of a year.

OCO-3 will dynamically control the viewing mode along each orbit via the PMA, with routine data collection consisting of nadir and glint measurements. Over land, where both nadir and glint measurements provide sufficient signal, the measurements will primarily be made in nadir mode, where both the optical path length and statistical probability of observing clouds are minimized.

Glint measurements are necessary over the ocean, as the surface reflectivity is not large enough to produce adequate signal, except in a few cases. A small offset from the true glint spot will be included to avoid saturation of the instrument. While the glint measurements provide larger signal over oceans, the longer optical path lengths and enlarged footprint of this geometry also make these measurements more sensitive to cloud cover (Miller et al., 2007).

The transition time for the PMA is required to be less than 50 seconds between nadir and glint modes, which translates into approximately 380 km along-track. In testing with the flight hardware, all moves were made within 10 seconds, which corresponds to about 75 km along track. Mission planning assumes the required 50 s move time, and thus, similarly to GOSAT, small land masses in the ocean will be measured in glint mode, while continental scale areas (areas that will be sampled for more than 200 seconds) will be measured in nadir mode. Unfortunately, this means that most inland fresh water bodies will be observed in nadir mode and will therefore not provide useful retrievals due to low signal to noise ratios. This will include bodies as large as, for example, the Great Lakes of North America. However, bodies of water as large as the Mediterranean Sea will be sampled in glint viewing. This is one of the key differences from OCO-2, where the measurement mode is specified orbit by orbit.

A subtlety of OCO-3 relative to OCO-2 is that, for nadir-land observations, the slit will remain perpendicular to the direction of flight since the instrument is not rotated to maintain measurements in the principle plane. This will produce a constant swath width of about 13 km, as depicted in Figure 1, which provides a best-guess representation of several frames viewing the Los Angeles metropolitan area. The ground footprint for glint mode measurements, however, will more closely resemble that of OCO-2, as the PMA will rotate to view near the specular reflection point.





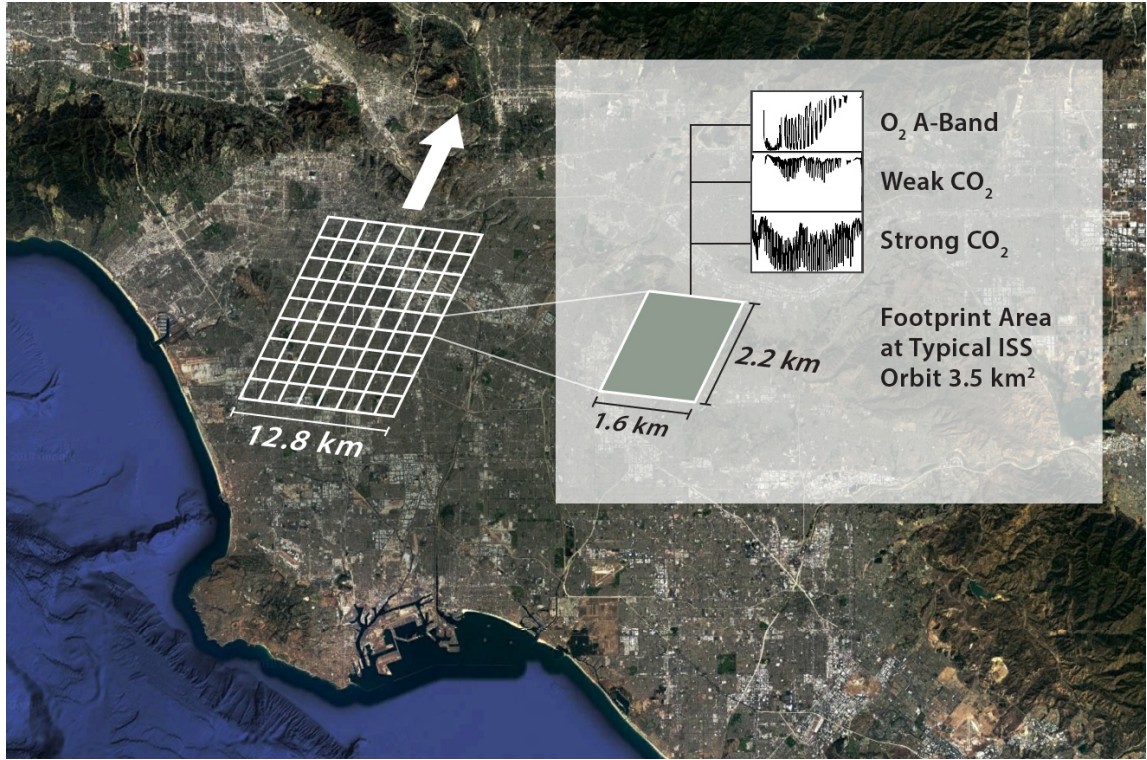

**Figure 1.** OCO-3 context image depicting 9 sequential frames acquired in 3.0 seconds in nadir viewing mode over the Los Angeles basin. Each frame contains 8 adjacent footprints in the cross-track direction of size 2.2 by 1.6 km, yielding a footprint area of approximately 3.5 km. Each footprint constitutes a sounding, containing high resolution spectra in the Oxygen-A, weak $CO_2$ and strong $CO_2$ bands. Image courtesy of Karen Yuen and Laura Generosa at JPL.

## 2.4 Target validation measurements

For the currently operating OCO-2, target mode measurements are taken over ground validation sites of the Total Carbon Column Observing Network (TCCON) (Wunch et al., 2010, 2011, 2017). The TCCON instruments are ground-based Fourier Transform Spectrometers that look directly at the sun (thus avoiding the complications of atmospheric scattering phenomena), and are used to derive total column carbon dioxide measurements with similar sensitivity as OCO-2 and OCO-3. The TCCON data are tied to the WMO scale for carbon dioxide through routine, ongoing overflights of aircraft equipped with in-situ sensors. The mechanics of the target mode observations of OCO-3 will be very similar to that of OCO-2, where data is collected using a sweeping, or dithering, pattern over the ground-based station. The width of the sampling area is determined by the combination of the instrument field-of-view and rotation of the footprints, which, in turn, is determined by the extent of PMA motion. Each target acquisition provides a set of overlapping observations that are used to statistically evaluate the retrieval performance for a range of viewing geometries as compared to the static TCCON ground-based measurement. The current OCO-2 mission





captures 1 or 2 target measurements per day, such that the total number gathered over the mission lifetime has been sufficient to perform validation (Wunch et al., 2017). OCO-3 will follow this basic strategy, although for some sites, where there are very few measurements in some seasons, e.g., at high latitudes in the winter, OCO-3 will potentially take more target measurements per day if it will improve the seasonal coverage for these sites.

## 2.5 Snapshot mode measurements

The agile pointing system of OCO-3 will also allow for collection of data in new spatial patterns relative to OCO-2. We have designed a snapshot mode, which can simply be thought of as a target observation with two dimensional sweeping, i.e., from side to side as well as back and forth. In this way, an area on the order of 80 km by 80 km can be sampled. The types of areas that will be sampled include $CO_2$ emission hotspots, terrestrial carbon focus areas, and volcanos. Based on analysis of fossil fuel emissions and uncertainties of the emissions estimates (Oda and Maksyutov, 2011; Oda et al., 2018), a nominal sampling strategy for the emission hotspots is being developed. Preliminary results suggest that 50 to 100 snapshots per day will be collected, consuming up to 200 of the approximately 650 day-light orbit minutes, i.e., 25 to 30% of the data volume. This will provide a novel data set for exploration by the scientific community that is focused on the remote sensing of greenhouse gases and SIF from space. The full details of the new snapshot mode will be presented in a future companion paper.

## 3 Simulated geometry, meteorology and L1b dataset

In this section, we discuss the simulation of OCO-3 data in terms of viewing geometry, meteorology and observed radiometric quantities such as data density and SNR, the latter of which is the primary driver of instrument precision. This will enable us to realistically discern the effects that the ISS orbit will have on our data products, in comparison to the sun-synchronous afternoon orbits of OCO-2. The generation of OCO-3 L1b radiances presented in this paper followed the same basic methodology as that used in previously published work on both GOSAT and OCO-2, e.g., (Bösch et al., 2006; O'Brien et al., 2009; O'Dell et al., 2012).

### 3.1 Simulated OCO-3 observation geometry

Actual ISS ephemeris data for the year 2015 were used to provide position and velocity vectors of the space station each second over the course of a year. To create a manageable data set for this work, samples were taken only once every 10 seconds, rather than at the true 3 Hz collection rate of the OCO-3 instrument. Also, only one sounding per frame, rather than eight, was used since the truth models lack the fidelity necessary for such high spatial resolution. The analysis presented in this work focuses on nadir and glint observation modes only, i.e., ignores transition, target and snapshot modes. This provides a baseline of the densest possible nadir and glint data if all the other viewing modes were disabled. As was mentioned in Sect. 2.5, it is estimated that as much as 25-30% of the data volume will be collected in snapshot mode, with some additional small amount (order of a few percent) collected in target mode.





Although the ISS latitude varies between $\pm 51.6°$, the OCO-3 PMA allows for measurements extending beyond this range to approximately $\pm 55.5°$ latitude. However, we found that useful measurements, i.e., those assigned a good $XCO_2$ quality flag as described in Sect. 4.3 and 5.3, are obtained at latitudes less than about 52°, where the solar zenith angle is less than about 73°. The exact range of latitudes measured on any given day will vary, depending on solar geometry and mechanical viewing

constraints from the ISS, as described below.

The top panels of Figure 2 show the number of measurements as a function of latitude and day of year, for the full annual data set. Nadir-land and glint-water observations are shown separately in the left and right columns, respectively. The data are binned in increments of 1 day and 2° latitude. The values in these figures, and in the accompanying discussion, must be inflated by 240 to reflect expected real sounding densities at the full spatiotemporal resolution. Note that the figures in this section use

L1b data collection density with no filtering. That is no cloud/aerosol prescreening or post L2 filtering have been performed here, topics which are discussed in later sections.

The most notable feature of both the nadir-land and glint-water observation density is the sinusoidal pattern with a period of about 70 days, yielding approximately 5 repeats per year. The nadir-land observation density ranges from close to zero soundings per bin below about 30° latitude (where there is little land), to about 25 soundings per bin (per day per 2° latitude)

up to about 20° N latitude. Above about 20° north latitude, the sampling density has significant latitude and time dependences, with a maximum of more than 300 soundings per bin at the northern extremity (approximately 55° N).

The density pattern for glint-water viewing is qualitatively very similar, but with much higher (50 to 60 soundings per bin) across most of the subtropics. The data densities can be over 300 soundings per bin at high latitudes near the satellite orbit inflection points. The simulated geometry used in this work takes into account the physical limitations of the PMA due to

interference from the solar panels and other constraints on the ISS. These physical restrictions have an especially large impact on the southern hemisphere glint data, as seen around DOY 120, 180 and 240.

The lower panels of Figure 2 show the same data subset to DOY range 60 to 119 (approximately March-April) to highlight the latitude and time dependence of the data collection across most of a single 70 day repeat cycle. Some interesting features, advantages and limitations of these collection patterns are presented after the discussion of the seasonal maps that are shown

next.

Another way to visualize the spatiotemporal distribution of the data is presented in Figure 3, which shows seasonal sounding density maps from the simulated data set, binned at 2° by 2° latitude. Here, and elsewhere in the manuscript, the seasons are defined as December/January/February (DJF), March/April/May (MAM), June/July/August (JJA), and September/October/November (SON). Here, no prescreening has been applied, and the reduction in spatiotemporal resolution requires inflation of the values

by 240 to get real expected sounding densities. There are just under a million soundings total for the full year, with approximately 25 soundings in each 2° bin over most of the globe per season. Presenting the data in this manner accentuates the high density of soundings at the orbit inflection points, although the drift in coverage with seasons is muted. The gaps in data collection so apparent in Figure 2 are no longer observed when the data have been aggregated monthly or seasonally. This has implications for the spatial and temporal scales of science questions that can be probed with the OCO-3 observations made

from the ISS.





Due to the precessing orbit, the local overpass time of the ISS ranges across all hours of the day. For OCO-3, with its pointing capability, this means that all daylight hours can in principle be sampled. Figure 4 uses Hovmoller diagrams to illustrate some of these features. The upper left panel shows the observation latitude as a function of hours from local noon (HFLN) and day of year (DOY) for the full annual data set. The dominance of the yellow colors suggests that a large fraction of the soundings

are taken at latitudes greater than 50° N.

The upper right panel shows the HFLN as a function of latitude and DOY for the full annual data set. Most of the observations are taken $\pm 5$ hours relative to local solar noon. Here the $\simeq 70$ day repeat cycle is evident, and the precession in observation time as a function of latitude becomes clear.

The lower panels of Figure 4 subset the data to DOY 60 to 119 (approximately March and April) to highlight some of the

detail across a single repeat cycle. Ten day periods are denoted with vertical lines in the diagrams. The data dropouts due to mechanical interference of the PMA by the ISS are seen at the higher southern latitudes. In general, the diurnal and spatial sampling pattern of OCO-3 aboard the ISS will vary significantly from the more familiar polar orbiting satellites. This will have implications on the $XCO_2$ and SIF science questions that can be explored.

An additional way to visualize the data is shown in Figure 5, which presents global maps of the sampling pattern for the six

sets of ten sequential days, highlighting both the spatial coverage and time of day sampling for a single repeat cycle. These maps clearly show the ascending/descending node variation in time and make it easy to comprehend the drift in HFLN as a function of day for any given location.

For example, imagine a field site of interest located in the Amazon basin. At the beginning of the repeat cycle this area is sampled in the ascending node (ground track orientatied from south-west to north-east) about 3-4 hours after local noon. Ten

days later (days 70-79), the orbits are still oriented in the ascending node, but now the sampling time has drifted closer to noon. The next ten day period (days 80-89) continues with ascending node orbits but sampling time about 3-4 hours before noon, when the biological processes driving SIF and $CO_2$ are only just ramping up for the day. Then suddenly, during the next 10 day period (days 90-99), the very same field site begins to be sampled in the descending node, i.e., from north-west to south-east, about 5 hours after noon. The sampling time again drifts closer to noon over the course of the next 10 days (days

100-109) before the final set of days (110-119) are observed in the morning hours. Disentangling the diurnal from the annual (and semi-annual) signals could be a challenge with this complex sampling pattern. Other sites at different latitudes will, of course, have different sampling patterns. The interpretation of this complex sampling pattern by global flux inversion models in an observing system simulation experiment (OSSE), as was performed for OCO-2 by Miller et al. (2007) and for GOSAT by Liu et al. (2014), is an interesting, but unexamined, issue that is outside of the scope of the current work.



**Figure 2.** Simulated sounding densities for nadir-land (left) and glint-water (right) for the annual (top) and DOY 60-119 (bottom) data sets. Data are binned in 1 day by 2° latitude increments. Values should be inflated by 240 to reflect real expected sounding densities at the full spatial (8 footprints per frame) and temporal (3 Hz) acquisition rates. To account for the large dynamic range, the density scale has been truncated at 150, although the extreme high latitudes contains up to 300 soundings per bin in some cases.

## 3.2 Simulated instrument polarization angle and Stokes coefficients

As unpolarized solar radiation traverses the earth's atmosphere (twice) prior to incidence upon the OCO instrument, interactions with particles, e.g., oxygen molecules and aerosols, as well as reflection off the surface, introduce some amount of polarization. Both the OCO-2 and OCO-3 instruments are sensitive only to radiation that is oriented perpendicular to the long axis of the



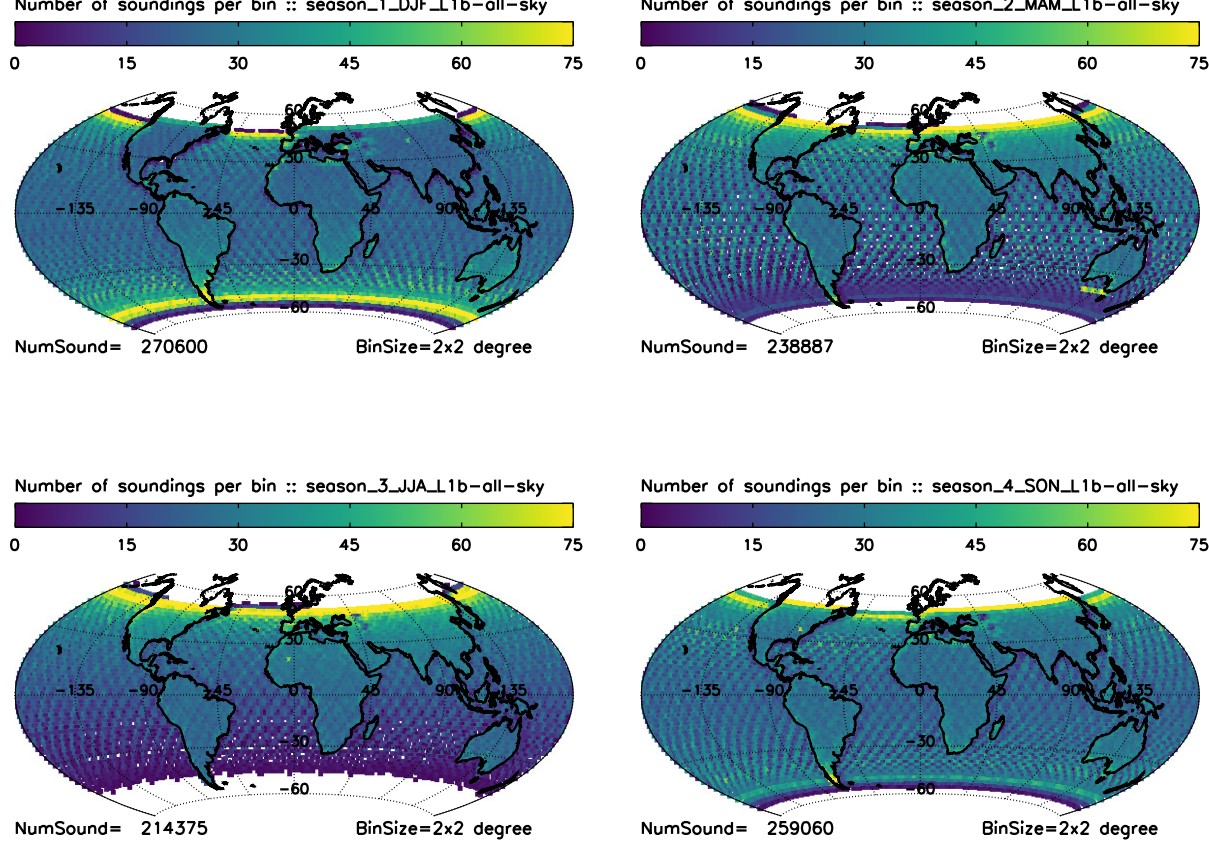

**Figure 3.** Seasonal L1b sounding density maps for 2° by 2° latitude bins. Values should be inflated by 240 to reflect real expected sounding densities at the full spatial (8 footprints per frame) and temporal (3 Hz) acquisition rates.

spectrometer slits [1]. This is critical over strongly polarizing water surfaces, but of lesser concern over land surfaces, which are only slightly polarizing. In the limit that the axis of accepted and actual polarization are perfectly orthogonal, the intensity observed by the instrument is identically zero. Although the degree (i.e., amount) of polarization of the reflected sun light is unknown (although somewhat predictable for water-glint measurements), the polarization angle of any particular sounding, which is a form of the throughput of the instrument, is a quantity that is calculable from illumination and observing geometries.

[1]As noted in (Crisp et al., 2017), the OCO-2 instrument was built erroneously; it was intended to be sensitive only to light parallel to the long axis of the spectrometer slits. OCO-3 was built in the same manner. This error was mitigated on OCO-2 by yawing the spacecraft in order to maximize the signal over ocean while simultaneously maintaining sufficient electrical power generated from sunlight on incident on the spacecraft solar panels. For OCO-3, electrical power comes from the ISS and is therefore a non-issue.



**Figure 4.** Hovmoller plots showing the observation latitude versus day of year and sampling time relative to local noon (top left) and the time relative to local noon versus DOY and latitude (top right) for the full year of simulations. Lower panels are subsets highlighting the patterns across days 60 to 119. Values should be inflated by 240 to reflect real expected sounding densities at the full spatial (8 footprints per frame) and temporal (3 Hz) acquisition rates.

The local meridian plane, formed by the local normal and the ray from the ground FOV to the satellite, forms the reference plane for polarization. The polarization angle of a measurement ($\phi_p$) is then defined as the angle between the axis of the instrument's accepted polarization and this reference plane (Boesch et al., 2015). Since fundamental physics predicts that scattered light will be preferentially polarized parallel to the plane of a horizontal surface, i.e., perpendicular to the reference plane, the larger (smaller) the polarization angle, the more (less) of the reflected sunlight incident on the instrument will pass



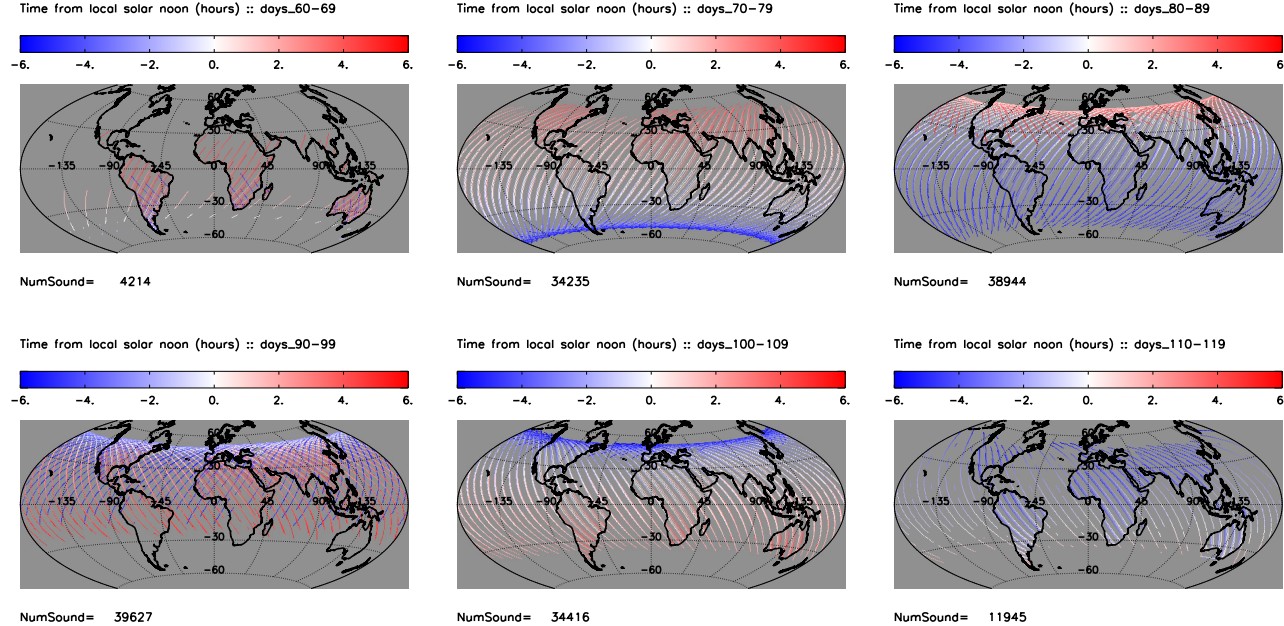

**Figure 5.** Simulated OCO-3 sampling pattern for six 10-day periods, colored by the time in hours relative to local noon. The pixel size of individual footprints has been magnified for viewing purposes.

through to the detectors, assuming a constant amount of polarization of the light. The geometry of individual soundings allows for the calculation of $\phi_p$, and hence to the calculation of the Stokes coefficients, which are given by the set of equations;

$$m_Q = \frac{1}{2} \cdot \cos(2\phi_p) \tag{1}$$

$$m_U = \frac{1}{2} \cdot \sin(2\phi_p) \tag{2}$$

5  A critical difference in OCO-3 observations aboard the ISS is that the polarization angle of the measurements will not be constant nor tied to latitude, as is the case for OCO-2, where $\phi_p$ is controlled by the dynamic orientation of the spacecraft to maximize instrument throughput (Crisp et al., 2017). For OCO-2 all data since November 2015 have been collected with the spacecraft yawed at 30°, resulting in a constant polarization angle of about the same sign and magnitude. For OCO-3, the polarization angle of the glint measurements will vary significantly in time and space as the pointing mirror is oriented to view

10  the ground target of interest.

Detailed optical modeling and laboratory tests have been performed to simulate the effects of the PMA induced changes to the polarization angle. The analysis showed that $\phi_p$ is a largely driven by the the PMA elevation angles with a some influence of the PMA azimuth angle. For elevation angles below 20°, the polarization angle is nearly equal to the elevation angle of



the PMA. In the nadir observing mode, the sensitivity to polarization is essentially negligible. However, for all off-nadir measurements, i.e., glint, transition, target, and snapshot modes, there will be a range of polarization angles, as the elevation angle of the PMA is adjusted to view the ground target. Ultimately this will have effects on the signal to noise ratios, as will be discussed in Section 5.1.

## 3.3 Simulated meteorology, gas and cloud/aerosol fields

In our simulations, vertical profiles of standard meteorological information needed to calculate realistic radiances were taken from the National Centers for Environmental Prediction (NCEP) (Saha et al., 2014). The NCEP database has a native spatial resolution of 2.5° latitude by 2.5° longitude (10,512 spatial points) with variables given on 17 vertical layers every 6 hours. For this work, the model was sampled at individual OCO-3 observations, defined by time, latitude, longitude, and surface elevation, for temperature, humidity, two meter temperature, surface pressure, and winds. The data are interpolated spatially and temporally to 26 vertical levels to create "scenes" for every individual OCO-3 sounding.

Vertical values of carbon dioxide for each sounding were sampled from the CarbonTracker 2015 database (CT2015) (Peters et al., 2007), with updates documented at http://carbontracker.noaa.gov), which has a native spatial resolution of 2.0° latitude by 3.0° longitude (10,800 spatial points), with $CO_2$ mole fractions given on 25 vertical layers every 3 hours. Data are interpolated in space and time to match individual OCO-3 soundings. Note that although the ISS ephemeris was taken from 2015, the CT database was sampled for 2012. Ultimately this makes no difference to the overall outcomes reported in this paper (which are not focused on actual carbon cycle science), but it is important to note that the simulations are representative of an earth-like system, not the actual conditions on Earth at the time of the soundings.

For each individual sounding, a cloud and aerosol profile containing 25 vertical layers was built based on a random selection from a monthly climatology of CALIOP profiles binned at 2.0° latitude by 2.0° longitude (16,200 spatial points), as described in the OCO simulator document (O'Brien et al., 2009). While these profiles do not capture the diurnal characteristics of cloud and aerosol fields (as CALIOP, flying in the A-Train, always has the same local overpass time), they are sufficient for this analysis, which is assessing statistics on monthly or seasonal time scales.

## 3.4 Simulated land surface model and SIF

A model of the earth's surface is a critical component for the calculation of reflected solar radiances. For land surfaces, scalar Bi-Directional Reflectance Distribution Functions (BRDF) were taken from the MODIS 16-day MCD43B1 product (Schaaf et al., 2002). For water surfaces, a fully polarized Cox and Munk model with a foam component based on wind speed was used. Additional details and citations can be found in the CSU simulator ATBD (O'Brien et al., 2009). Realistic estimates of solar induced chlorophyll fluorescence (SIF) from biological activity were added to the oxygen A-band L1b radiances based on the implementation of Frankenberg et al. (2012). A static gross primary production (GPP) climatology of Beer et al. (2010), which is a mean monthly climatology based on the 18 IGBP surface types at 0.5° by 0.5° latitude and longitude resolution, is scaled to a daily average SIF value using the empirical scaling factor of Frankenberg et al. (2011). Daily average SIF is converted to instantaneous SIF via scaling by the instantaneous solar insolation relative to the average for that day and location. The




wavelength dependence is a double Gaussian function as given in Frankenberg et al. (2012). Overall, this provides values of SIF that are representative in time (seasonal and diurnal cycle) and space (as a function of latitude and local plant physiology). It is worth noting that the use of the static GPP climatology does not allow for interannual variability, but this has no effect on the single year of simulated data presented here.

## 3.5 Simulated L1b radiances

Radiances, as would be observed by the OCO-3 instrument in space, are calculated using the same forward model (FM) that has previously been employed for GOSAT and OCO-2 simulation studies, e.g., O'Dell et al. (2012). The FM consists of a atmospheric model, surface model, instrument model, solar model, and radiative transfer model.

The solar spectrum is comprised of two parts; a pseudo-transmittance spectrum (Toon et al., 1999) and a solar continuum spectrum (Thuillier et al., 2003), used to produce a high-resolution, absolutely-calibrated input solar spectrum for the forward model (Boesch et al., 2015). For this work, the gas absorption coefficients, i.e., spectroscopy, of the current operational OCO-2 B8 L2 algorithm, ABSCO v5.0.0, were used. The instrument model, which includes the instrument line shapes (ILS), radiometric characteristics, polarization sensitivity, and noise specifications, were taken from the OCO-3 thermal vacuum tests performed in September 2016. Noise was applied to the calculated radiances via the same model used for OCO-2, as described in Rosenberg et al. (2017). The radiative transfer calculation accurately accounts for multiple scattering from clouds and aerosols as well as polarization, as described in O'Brien et al. (2009) and references therein.

## 4 Level 2 preprocessors and full physics retrieval algorithms

The primary data products for OCO-2 and OCO-3 are the column-averaged dry-air mole fraction of $CO_2$ ($XCO_2$) and the solar induced chlorophyll fluorescence (SIF), both of which can be used to help constrain the global carbon cycle, e.g. Eldering et al. (2017b); Sun et al. (2017). For this work, the simulated L1b radiances were analyzed with the same tools used in OCO-2 operational data processing, as described in Section 4 of Eldering et al. (2017a). These steps include prescreening, L2 retrievals, quality screening and the application of a bias correction for $XCO_2$. This section briefly discusses each of the components as it relates specifically to the OCO-3 simulations. Relevant citations containing the full details are provided.

### 4.1 Preprocessors

Cloud screening was performed using only the A-Band Preprocessor (ABP), as described in Taylor et al. (2016). The ABP identifies cloud contaminated soundings primarily via a threshold on the difference in retrieved and prior surface pressure in the Oxygen-A band, typically $\pm 25$ hPa. Although operational OCO-2 data also utilizes a weak filter on the ratio of $CO_2$ retrieved independently in the strong and weak $CO_2$ bands by the IMAP-DOAS Preprocessor (IDP), we did not implement this filter for cloud screening. The IDP $CO_2$ and $H_2O$ ratios were however used for post L2 retrieval quality filtering and bias correction. In addition IDP performs a retrieval of SIF, which is used as a prior for the full physics L2 SIF retrieval. After some additional post processing, such as removing a zero level offset (ZLO), the IDP SIF becomes the formal LiteSIF product that is



available on the NASA DISC. Both preprocessors neglect scattering in the atmosphere (except Rayleigh scattering is included in ABP), making them computationally very efficient.

## 4.2 Full physics retrieval algorithm for $XCO_2$ and SIF

The soundings that were identified as clear by the ABP cloud flag were then run thru the OCO-2 B8 operational retrieval

algorithm. The algorithm was first described in Bösch et al. (2006) and Connor et al. (2008) prior to the failed launch of OCO-1 in February 2009, and was later applied to GOSAT as described in O'Dell et al. (2012). Recent updates and a complete description of the modern B8 version of the algorithm can be found in Boesch et al. (2015) and O'Dell et al. (2018).

In summary, the algorithm is an optimal estimation retrieval with a prior that minimizes radiance residuals, i.e., chi-squared, to maximize the a posterior probability. The solution is solved on 20 vertical levels, with the state vector containing $CO_2$ dry

air mole fraction, aerosol parameters, surface albedo, wind speed, water vapor, and a temperature scaling factor. The high spectral resolution measurements of top of the atmosphere reflected radiances measured by sensors such as GOSAT, OCO-2, or OCO-3 serve as the primary source of information in the retrieval. The measurements are coupled with an a priori state of the atmosphere in order to constrain the inversion. Within the L2 retrieval, modeled spectra are generated by a radiative transfer code as described in O'Dell et al. (2012) and Boesch et al. (2015). Although they share many components, the L2 code

base differs slightly from the RT FM used in the generation of the L1b radiances, thus creating a realistic error source in the simulation exercise, i.e., imperfect radiative transfer.

## 4.3 Filtering and bias correction approach

The NASA operational procedure for both OCO-2 and GOSAT applies a quality filtering (QF) and bias correction (BC) process to the retrieved $XCO_2$ (O'Dell et al., 2018). Correlations between variables and $XCO_2$ error variability are quantified and used

to develop the filtering thresholds and linear bias correction equations. The quality filtering is designed to remove soundings with anomalous $XCO_2$ values relative to other soundings in close proximity, making use of the assumption that real variations in $XCO_2$ are quite small ($<1$ ppm) on small scales ($<100$ km). Some form of "truth" metric, or truth proxy, is required with which to calculate an "error" in $XCO_2$. For operational OCO-2 data, several forms of a truth proxy are used, as detailed in Section 4.1 of O'Dell et al. (2018).

A similar treatment was applied to the OCO-3 simulation dataset. However, with simulations it was possible to use the actual true $XCO_2$ as the truth proxy in the QF and BC procedures. There are both advantages and disadvantages to the circularity imposed by knowing the true values of the atmospheric state. In this case, we expect that the use of the truth data will result in an overly optimistic QF and BC.

At its completion, the QF/BC procedure assigns to every sounding a binary flag indicating good/bad quality, as well as a BC

value (in units ppm). The operational OCO-2 BC equation contains three components; a correction based on retrieval variables (parametric), a correction for inter-footprint dependence and a global bias, each calculated separately for land (combined nadir and glint) and ocean-glint. For the OCO-3 simulations the inter-footprint bias is not needed since only a single footprint per



frame was calculated. Explicit results from the procedure as performed on the OCO-3 simulations are given in Sections 5.3 and 5.4.

## 5   Results

This section discusses characteristics of the L1b radiances, performance of the preprocessors, and application of the quality

filter and bias correction methodology before presenting the L2 $XCO_2$ results. In addition, we provide a brief analysis of the SIF determined by the IDP retrieval. Table 1 summarizes the number of soundings in the simulated data set at each stage of the analysis, broken out by nadir-land and glint-water observations.

### 5.1   Simulated L1b radiance characteristics

At a gross level, the characteristics of the simulated OCO-3 radiances are very similar to those from real OCO-2 measurements.

The high resolution spectra for OCO-3 (not shown) exhibit the expected absorption features that allow for cloud and aerosol screening, and the retrieval of surface pressure and SIF (from the $O_2$ A-band) and $XCO_2$ (from the weak and strong $CO_2$ bands).

However, some differences are expected between the two sensors in both measured signal and instrument noise due to the addition of the PMA and calibration characteristics of the spectrometers, e.g., dark noise, stray light and ILS. Optical

inefficiencies in the OCO -3 PMA will reduce the transmission of light by about 17% in the $O_2$ A-band, and 7% and 5% in the weak and strong $CO_2$ bands, respectively. To compensate for the effects of the PMA, the instrument aperture of the $O_2$ A-band was increased. When all of the optical elements and instrument changes are considered, the $O_2$ A-band transmission of OCO-3 will be about 95% of OCO-2, while the weak and strong $CO_2$ bands will have 75% of the transmission of OCO-2, thus reducing the observed signal for the same scene.

The instrument calibration parameters for the OCO-3 simulations reported here were taken from results of the September 2016 pre-launch thermal vacuum testing (TVAC), which was performed using an early version of the instrument telescope and without the PMA installed. The noise coefficients were adjusted post-hoc to account for the reduced optical throughput caused by the PMA. Although the final round of pre-launch TVAC calibration was completed in May 2018, the results were still not available at the time of writing. However, based on preliminary analysis, the updated instrument characteristics are not

expected to change at a level that would greatly effect the results presented here.

A key characteristic of the radiances measured by satellite sensors is the SNR, which effectively determines the information content of the measurements, thereby controlling the precision of the retrieval estimates of $XCO_2$ and SIF. The signal for each band is calculated from continuum level radiances, using the ten channels with the highest values, after filtering for outliers that occasionally exist due to cosmic rays or some other random electronic anomoly. The OCO noise model combines contributions

from a constant background (dark noise) term and a photon (shot noise) term, the later of which is proportional to the square root of the radiance (Rosenberg et al., 2017).





Figure 6 compares the OCO-2 SNR calculated from the operational noise model (solid traces) against OCO-3 (dashed traces) versus a measure of the surface brightness (the albedo scaled by the cosine of the solar zenith angle, $A \cdot \cos(SZA)$). The left panel displays the SNR of each spectral band for both sensors, while the right panel shows the ratio of the two sensor's SNR per spectral band. This data demonstrates that the only situation in which OCO-3 has a higher SNR than OCO-2 (values $>1.0$ on the right panel) is in the $O_2$ A-band for $A \cdot \cos(SZA) \gtrsim 0.15$. This typically occurs over very bright deserts and during glint-water measurements when the sun is low in the sky. It is worth noting that the $O_2$ A-band is used primarily for cloud and aerosol detection and for the L2 FP surface pressure retrieval as well as for SIF.

In both of the weak and strong $CO_2$ bands (green and red, respectively, in the figure), OCO-3 always has a significantly lower SNR than OCO-2. This reduced SNR can be attributed to both increased noise due to the use of noisier instrument detectors that were spare parts from the rebuild of the OCO-2 instrument, and to decreased signal incurred by the use of the PMA, a polarizer in the telescope, and a larger center obscuration in the entrance optics.

The overall SNR differences are captured in the histograms of Figure 7, which compare the OCO-3 simulations with the real SNR for operational OCO-2 B8 measurements acquired in 2016. The operational OCO-2 data has been down selected to include only a single footprint and one sounding every 10 seconds to provide a fairer comparison against the OCO-3 simulations. Both data sets have been screened using the L2 FP quality flags, which were introduced in Section 4.3, and will be discussed in more detail in Section 5.3. At a gross level, the data look reasonably similar, although a few key distinctions stand out, particularly that the slightly brighter OCO-3 $O_2$ A-band is primarily due to a long tail of high values for glint-water soundings. The OCO-3 weak $CO_2$ band exhibits a substantially lower SNR for glint-water compared to OCO-2, while the strong $CO_2$ band tends to be also lower than OCO-2. These figures show that the OCO-3 data will include less data with SNR values over 600, and more data with SNR between 200 and 400. Previous OCO-2 studies and experience with the real data show that an SNR of 200 is sufficient to achieve the desired precision of the retrieval algorithm. As will be shown in Sections 5.5 and 5.6, the L2 FP retrieval still provides good estimates of $XCO_2$ and SIF on this set of OCO-3 simulated radiances, even with the lower SNR values.

Maps comparing the simulated OCO-3 SNR to the operational B8 OCO-2 data for each spectral band are shown in Figure 8 for the month of April. Qualitatively, the overall patterns agree quite well, although the difference in latitudinal coverage from the two spacecraft is noteworthy. We also note that a higher fraction of the soundings that converge in the L2 FP retrieval are assigned a good quality filter in the OCO-3 simulations (approximately 70%), versus only about 40% for real OCO-2 B8 data. This is likely driven by deficiencies in the simulation setup, such as lack of a Southern Atlantic Anomaly model, and a parameterized cloud and aerosol scheme in the L1b simulations that lacks full realism. We expect that real on-orbit OCO-3 good quality sounding fractions will in reality be closer to the OCO-2 values.

For both sensors, the highest SNR's are obtained over un-vegetated (bright) land, and for glint-water when the sun is low in the sky, producing a strong specular reflection. The lowest values of SNR occur when when the sun is high in the sky, and for vegetated (dark) land surfaces at higher latitudes. As with OCO-2, the weak $CO_2$ band displays the highest SNR values, while the $O_2$ A-band and strong $CO_2$ bands have lower but comparable SNRs.



A final glimpse of the SNR characteristics are shown in Figures 9 and 10, which compare the SNR dependence on latitude and SZA for both sensors. The restriction of OCO-3 to latitudes less than approximately 54° is pronounced, especially for the glint-water soundings, when comparing to the wider latitudinal distribution obtained from OCO-2. This is simply a consequence of the ISS precessing orbit versus the polar orbit of OCO-2. On the other hand, the OCO-3 measurements span SZA's from approximately 85 to 0°, while OCO-2 measurements are limited to a minimum SZA of approximately 20° in the subtropics and larger than 30° above 40° latitude. As was demonstrated previously in the histogram plots (Figure 7, we again see here that for nadir-land the OCO-3 SNR values tend to be lower for OCO-3 compared to OCO-2 in all spectral bands, with the exception of a few high $O_2$ A-band SNRs around 20° latitude which correspond to the Sahara desert. For glint-water soundings, there is a population of very high SNR values ($> 800$) spanning the full latitudinal space, at SZA around 60° due to the very bright specular glint spot achieved under these conditions.

While real on-orbit SNR characteristics will likely differ somewhat from those shown here, these simulations suggest that the instrument has been well built and well calibrated and should provide SNR that meets the mission requirements. In addition, due to the nature of the precessing orbit of the ISS, we expect that the SNR distribution, which fundamentally drives the information content in the L2 FP retrievals, will not be tied to latitude in the same way that it is for OCO-2. This has implications as to the spatial patters of good quality $XCO_2$ and SIF retrievals, as will be discussed in the following sections.





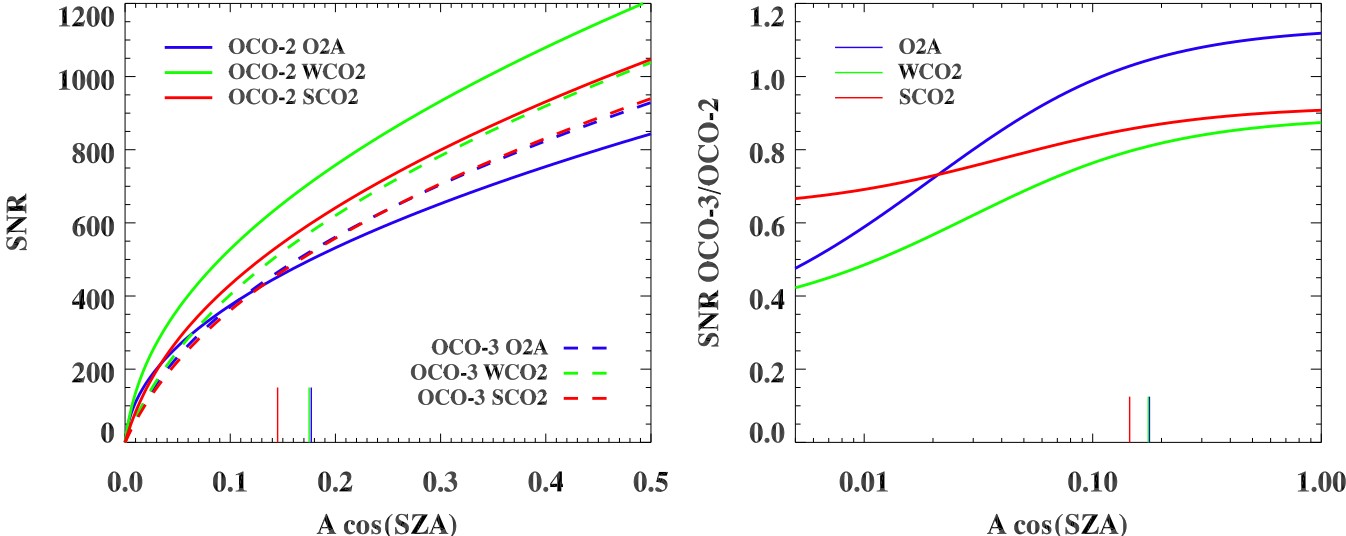

**Figure 6.** OCO-2 and OCO-3 mean SNR (averaged across footprints and channels) for each band as a function of the product of surface albedo and cosine of solar zenith angle (left panel). The quantity $A \cos(\text{SZA})$ is proportional to the reflected sunlight off of a surface. The right panel shows the ratio of OCO-3 SNR to OCO-2 SNR using a logarithmic abscissa scale. The small vertical lines represent $A \cos(\text{SZA})$ for Railroad Valley at the winter solstice. OCO-3 SNR is lower at lower signal levels because it has a higher noise floor than OCO-2.

## 5.2 Preprocessor performance

For this simulation experiment only the ABP cloud flag was used to select soundings, although real operational sounding selection is expected to be slightly more elaborate (see Sec. 2 of O'Dell et al. (2018)). In particular, no IDP variables were used in the L2 sounding selection here, although they were used later in the post-filtering and bias correction. The results shown in Table 1 indicate that about a quarter (24.2%) of all of the observations passed the ABP cloud flag, leaving about 250,000 to run through the L2 FP retrieval. Broken out by viewing mode, approximately one third (31.8%) of the nadir-land and one fifth (20.2%) of the glint-water observations passed the ABP cloud flag. These statistics are roughly similar to those seen in real OCO-2 operational processing.

Figure 11 shows maps of the clear-sky fractions in each 2° spatial bin (left) and the resulting clear-sky sounding densities (right). As expected, the highest fraction (up to about 75%) of the scenes pass in the arid land regions, where there are few clouds and aerosols. The southern subtropical oceans also tend to have areas of moderately high passing rates around 50%. Tropical land regions, e.g., the Amazon, Congo and Indonesian rainforests, have on average only about 5 to 10% passing rates. Most temperate land regions such as the eastern United States and southern Europe generally have passing rates of about 30%.







**Figure 7.** SNR histograms comparing simulated OCO-3 (left column) to operational B8 OCO-2 (right column) for nadir-land (top row) and glint-water (bottom row). The colors represent the three spectral bands, as described in the legend. Both data sets have been filtered using their respective L2 quality flags. The median value for each spectral band is shown as a vertical dashed line in the corresponding color.

These results meet expectations and are qualitatively very similar to those seen in Fig. 1 of O'Dell et al. (2018) for OCO-2 operational B8 data.

The right panel of Figure 11 confirms that the highest density of cloud-free soundings (more than 100 per 2° bin) are found over the arid regions of the globe, as expected. In addition, a large number of soundings are found over the northern hemisphere land at the satellite orbit inflection points. Much of the temperate land regions contain about 30-50 soundings per bin, while few soundings remain over tropical forests. In glint-water viewing, the regions of high clear-sky fraction have about 50 to 80 soundings per 2° bin, while the cloudy areas contain only about 10 soundings per bin selected for processing by the L2 FP





retrieval. Recall that on-orbit OCO-3 sounding densities will be about 240 times greater due to the reduced spatiotemporal sampling used in this simulation set.



### 5.3 Application of XCO$_2$ quality filter

The L2 FP retrieval algorithm described in Section 4.2 was applied to the cloud screened set of soundings, and then, as with operational OCO-2 data, a set of post-processing filters were implemented to determine the binary XCO$_2$ quality flags (QF). Details of the methodology are documented in O'Dell et al. (2018). Here, the true XCO$_2$ for each sounding was used as the truth metric to assess residual biases and errors. This provides perhaps an overly optimistic interpretation of the results, and should be considered an upper limit on the actual performance expected from the real system.

Explicit values of the QF thresholds determined for the OCO-3 simulations are presented in Tables 2 and 3. The QF methodology was applied independently to the nadir-land and glint-water scenes, as is done with real OCO-2 data. Eleven variables were used to form the QF for nadir-land, while nine were used for glint-water. Not surprisingly, many of the same variables are selected for quality filtering the OCO-3 simulations as were used in the operational OCO-2 procedure. See Figures 10 and 11 of O'Dell et al. (2018). Approximately 70% of the soundings that converged in L2 FP were assigned a good quality flag.

The quality filtering process had similar impacts on data volume across all months (not shown). On average, global data densities of good QF soundings in the simulations were 11,000 to 12,000 soundings per month, or 33,000 to 36,000 per season. When using the full spatiotemporal resolution, this translates to approximately 2.5 million soundings per month (7.5 million per season), similar to the density of OCO-2 B8 data.

Figure 12 shows seasonal plots of the fraction (left column) and number (right column) of soundings passing the quality filters for each season (DJF, MAM, JJA, SON), binned in 4° degree lat/lon bins. The spatial patterns are useful, but the absolute numbers need to be inflated by 240 to reflect actual predicted on-orbit throughput. These maps can be compared with those shown in Figure 12 of O'Dell et al. (2018).

In general, the QF throughputs for glint-water are quite high (>70%) in the tropics and subtropics (<30° latitude), and display very little seasonal cycle. The QF throughput is persistently low for glint-water observations at the extreme latitudes. The QF throughputs are more varied for nadir-land observations, and a modest seasonal cycle is seen for some regions. But overall, the results look qualitatively similar to those from OCO-2 for the B8 operational data set, and demonstrate that the methodology is a robust procedure.





### 5.4 Bias correction of $XCO_2$

The final bias correction (BC) incorporates four of the QF variables for nadir-land and three for glint-water as shown in Table 4. There are notable similarities and differences in the selected variables when comparing between the OCO-3 simulations and real OCO-2 data. See Section 4.3.1 in O'Dell et al. (2018).

Figure 13 illustrates how the final BC parameters for land affect the $XCO_2$ error. Each panel shows median binned values of the $XCO_2$ error (retrieved minus true in ppm) versus a particular retrieval variable shown by the heavy black dots. Also shown are the range in $XCO_2$ error (thin vertical bars) and the least squares linear fit (thin dashed line). To provide context, the relative histogram of points is shown in the background by the shaded grey region. The slope of the fit, the standard deviation of the $XCO_2$ error post BC, and the percent of the variance explained by this variable are given in the legend. The original standard
deviation is shown in the upper left panel for reference.

For land, 25% of the variance is explained by the L2 dP (retrieved surface pressure minus a priori), while another 15% is explained by the combined retrieved AOD from dust and sea salt aerosols. An additional 3% and 2% are explained by the L2 fine mode AOD and the water vapor scaling factor, respectively. We believe that a minor indexing bug found in the meteorology is responsible for the reliance on water vapor. The final reduction in $XCO_2$ error is shown in Table 6, which gives the standard
deviations (sigma) in the retrieved $XCO_2$ with and without QF and BC. For land, sigma was reduced from 1.88 ppm to 0.85 ppm after application of both QF and BC.

Figure 14 is similar to Figure 13, but for the glint-water scenes. Here, 18% of the variance in $XCO_2$ error is explained by the IDP $CO_2$ ratio, while another 16% is explained by the ABP dP. An additional 7% is explained by the L2 dP. As seen in Table 6, sigma was reduced from 2.15 ppm to 0.52 ppm for glint water soundings after application of both QF and BC.

Spatial seasonal maps of the total bias correction (in units ppm) are shown in Figure 15. Although the results are qualitatively different from those seen for operational OCO-2 B8 data presented in O'Dell et al. (2018), this follows expectations in that here we are working with simulated data which is more internally consistent then real data, especially with respect to ABSCO and meteorology. These results underscore the conclusion that even given nearly perfect alignment of the retrieval model with the truth, there are still retrieval errors that induce biases and scatter into the estimates of $XCO_2$. This is particularly true of
aerosols, which are a continued known source of trouble in virtually all retrievals of greenhouse gases from space, e.g., (Aben et al., 2007; Butz et al., 2009).





### 5.5 Retrieved XCO$_2$ characteristics after filtering and bias correction

One of the objectives of this study was to analyze the error on the retrieved XCO$_2$ from OCO-3. Here the "actual" error is given as the retrieved value minus the known truth (after applying the averaging kernel correction), and is denoted $\Delta$XCO$_2$. The "predicted" error is an L2 FP retrieval state vector parameter that provides the theoretical error due to the combination

of measurement noise plus smoothing and interference errors, as discussed in Boesch et al. (2015). The actual errors in the simulated framework are expected to be substantially lower than those seen in OCO-2 operational data, while the predicted errors should be roughly equivalent due to use of similar instrument model and retrieval algorithm.

As demonstrated in Figure 16, which shows the histograms of $\Delta$XCO$_2$ for nadir-land and glint-water data separately, $\Delta$XCO$_2$ is small and effectively corrected by the filtering and bias correction process on an annual average basis. Overall,

QF/BC reduces the median $\Delta$XCO$_2$ bias from -0.12 to -0.02 ppm for land soundings and from 0.23 to -0.09 ppm for glint-water soundings. The histograms indicate that the filtering process identifies a significant population of glint-water soundings with large negative biases up to about -12 ppm.

The seasonal spatial distributions of $\Delta$XCO$_2$ are shown in the maps of Figure 17. These can be compared to Figure 19 of O'Dell et al. (2018). While the qualitative patterns of actual XCO$_2$ errors are quite different between OCO-2 B8 and simulated

OCO-3 data, note that the dynamic range of the scale is much lower for OCO-3 ($\pm 1$ ppm) compared to OCO-2 ($\pm 3$ ppm). Again, this follows expectations since the truth proxy for the simulations is the actual truth, while that metric is not available in the real world.

For the OCO-3 simulations, after QF/BC have been applied, the errors are largely uncorrelated with any geophysical or retrieval parameters. Specifically, we used the glint-water soundings to check for correlation of both the raw and BC XCO$_2$

data against latitude, solar zenith angle, polarization angle, SNR (per spectral band), and the true aerosol optical depth. The results are summarized in Table 5. It is worth noting that the true AOD was not used as a bias fitting parameter, yet there is a high reduction in the correlation with $\Delta$XCO$_2$. The very small slopes, offsets and linear correlation coefficients that remain after application of the QF/BC indicates that remaining errors in the XCO$_2$ are likely driven by retrieval errors such as the aerosol parameterization (Nelson et al., 2016).

Shown in Figure 18 are the OCO-3 actual ($\Delta$XCO$_2$) versus the L2 FP retrieval predicted XCO$_2$ errors comparing the unfiltered raw, the filtered raw and the filtered and bias corrected data. There is a huge improvement in the performance after filtering is applied. Additional improvements are achieved by application of the bias correction. Results fall nearly on the one-to-one line, with some exception for nadir-land soundings when the predicted error falls below about 0.7 ppm, in which case the actual error is larger than theory. Overall, these results provide evidence that the filtering and bias correction methodology

is a robust procedure that performs according to theory given a (nearly) perfect truth metric.





## 5.6 Retrieved SIF characteristics

Similar to the analysis of $XCO_2$ in the previous section, here we present the IDP SIF relative to the true values and examine both the actual and predicted errors. In order to calculate the "actual" SIF error (retrieved - true), the L1b truth values, which are calculated at 755 nm in the simulator code, were first wavelength shifted to match the IDP retrieved values at 758.65 nm and 769.95 nm. Note that these IDP channel values denote the center points of the retrieval ranges but are labeled as 757 nm and 771 nm throughout the code and analysis for historical purposes as reported in Section 2.2 of Sun et al. (2018). For brevity, we only show results for the 757 nm band, although there is no reason to expect a significant difference in performance in the 771 nm band.

The retrieval of SIF from space is highly sensitive to measurement error, i.e., instrument noise (Frankenberg et al., 2014). It is therefore a common practice to aggregate some number of soundings, N, in order to minimize the random noise. Since these simulations have spatiotemporal sampling of 1/240 of the real expected value, we "noise-corrected" our results by scaling the noise as;

$$\text{SIF}' = \text{SIF} + \frac{\text{noise}}{\sqrt{240}}, \tag{3}$$

where SIF represents the retrieved values using noiseless radiances, and noise is calculated by differencing the with- and without- instrument noise retrievals.

Unlike with $XCO_2$, the retrieval of SIF using the solar Fraunhofer lines is not highly sensitive to cloud and aerosol contamination (Frankenberg et al., 2012) We therefore did not apply any strict prescreening or L2 quality flagging prior to running the IDP retrieval on the L1b files. Although IDP can in principle retrieve SIF over water, glint-water soundings were ignored since the L1b simulator assumes zero SIF in these cases. Out of the 337,000 total land soundings approximately 12% of the IDP retrievals failed altogether (identified by way of fill values in the output).

We then applied a post processing quality filter on the approximately 300,000 successful nadir-land soundings, which included removal of scenes with SZA > 70, for which the actual SIF error became very large. We also removed 49 soundings where the predicted retrieval noise as a function of the continuum level radiance fell well outside a smooth fitting criteria. For unknown reason these small number of soundings had very large actual SIF errors. After application of the filtering criteria, approximately 264,000 land soundings (88% of the successful soundings [2]) remained in the annual data set.

The top row of Figure 19 shows maps of the true (left) and IDP retrieved SIF (right) for the 757 nm band for the JJA season, when the northern hemisphere land photosynthetic activity is at its annual maximum. The units are expressed in radiance space as Watts-per-square-meter-per-micron-per-steradian ($W/m^2/\mu m/sr$). SIF typically comprises at the maximum about 1 to 2% of the total radiance measured at the top of the atmosphere by satellite sensors (Frankenberg et al., 2012). At a gross scale, the true and retrieved values show the expected patterns, with SIF up to about $1.2\,W/m^2/\mu m/sr$ in densely vegetated tropical regions, when aggregated to 1° by 1° bins, and (near) zero SIF over barren deserts, high mountains and high latitudes.

---

[2]It is coincidence that both the fraction of soundings that failed the IDP retrieval and the number that were flagged by our post processing filter is 12%.





Although they appear qualitatively similar, the absolute difference in true and retrieved values suggests that the IDP tends to underestimate SIF. This is particularly so for higher fluorescing areas, as seen in the middle left of Figure 19. To better quantify the differences in the retrieved and true values, a fractional difference was calculated after masking out the soundings with true SIF less than $0.2\,\mathrm{W/m^2/\mu m/sr}$ (to avoid the intractable math of the ratio of two numbers close to zero). As seen in the middle right panel, the median fractional differences in SIF is -9% for this subset of the data, with individual soundings having outliers as large as -240%. The bottom row of panels shows the correlation between the true and retrieved values for the JJA data, with and without the true SIF $< 0.2\,\mathrm{W/m^2/\mu m/sr}$. The Pearson linear correlation coefficients are very close to 1, indicating that the retrieval of SIF from the IDP is expected to perform well for the OCO-3 instrument, as has already been shown for operational OCO-2 data (Sun et al., 2017, 2018).

The comparison between the "actual" (retrieved minus true) and "predicted" SIF error are shown in Figure 20. Here we show both the unfiltered, i.e., no quality flag applied, and filtered annual data sets. When the retrieval predicted error is above approximately $0.25\,\mathrm{W/m^2/\mu m/sr}$ the actual error is in very close agreement for the filtered data. Below predicted error of $\simeq 0.25\,\mathrm{W/m^2/\mu m/sr}$, the filtered data set still tends to have a slightly larger value of actual error. For predicted error $< 0.25\,\mathrm{W/m^2/\mu m/sr}$ the actual error becomes quite large in the unfiltered data set.

A comparison of the single sounding SIF precision between the OCO-3 simulations and the operational B8 OCO-2 data is given in Figure 21 for both the 757 nm and 771 nm windows. For both instruments, the precision is an increasing function of the continuum level radiance, as explained in Section 3.1 of Frankenberg et al. (2014) in association with Figure 8. The darker the scene, the better the precision, due to decreasing noise in the Fraunhofer lines. Overall, both instruments have better precision at the shorter wavelength channel. This analysis suggests that OCO-3 SIF precision will be 10-20% worse than for OCO-2, which may be directly ascribable to the noisier instrument detectors. More research and analysis on IDP SIF using real on-orbit measurements will be needed to determine answers.





**Figure 8.** Maps comparing the SNR of OCO-3 (left) to OCO-2 (right) for each spectral band (rows) for the month of April binned in 2°
latitude bins. Both data sets have been filtered using the L2 quality flag. The operational OCO-2 data has been downselected to include a
single footprint and one sounding every 10 seconds to provide a fairer comparison against the OCO-3 simulations. The OCO-2 data also
includes both nadir and glint land soundings, in addition to glint-water.







**Figure 9.** Comparison of the nadir-land SNR of OCO-3 (left) to OCO-2 (right) for each spectral band (rows) for the full annual data set. Both data sets have been filtered using the L2 quality flag. The operational OCO-2 data has been downselected to include a single footprint and one sounding every 10 seconds to provide a fairer comparison against the OCO-3 simulations.



**Figure 10.** Same as Figure 9, but for glint-water soundings.





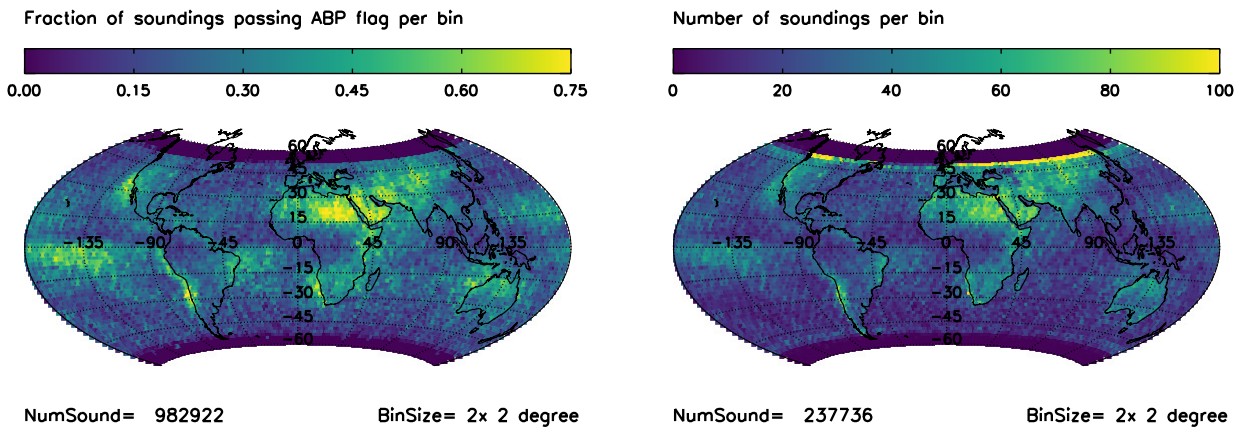

**Figure 11.** Maps of the annual fraction of soundings passing the ABP cloud flag (left) and the resultant clear-sky sounding density (right) binned $2°x2°$ latitude. Number densities should be inflated by 240 to provide real estimated number of soundings at the full spatiotemporal sampling.





**Figure 12.** Maps of the seasonal throughput (left column) and the resulting sounding densities (right column) in 4° lat/lon bins after application of the L2 FP quality flag. Inflate densities by 240 to account for on-orbit spatiotemporal sampling.





**Figure 13.** Final bias correction variables used for land scenes illustrating the correlation of the actual error ($\Delta XCO_2$ defined as retrieved - true) as a function of variable value. The leading retrieval parameters that explain the maximum variance for land are the L2 delta surface pressure, L2 $H_2O$ scale factor, and two L2 aerosol terms. The original standard deviation ($\sigma$) of the data set is given in the upper part of the first panel, with the cumulative reduction in $\sigma$ and percent variance explained given in the lower right.





**Figure 14.** Same as in Fig.13, but for water scenes. Here the leading retrieval variables that explain the maximum variance are the L2 delta surface pressure, the IDP co2_ratio and the ABP dp_cld.





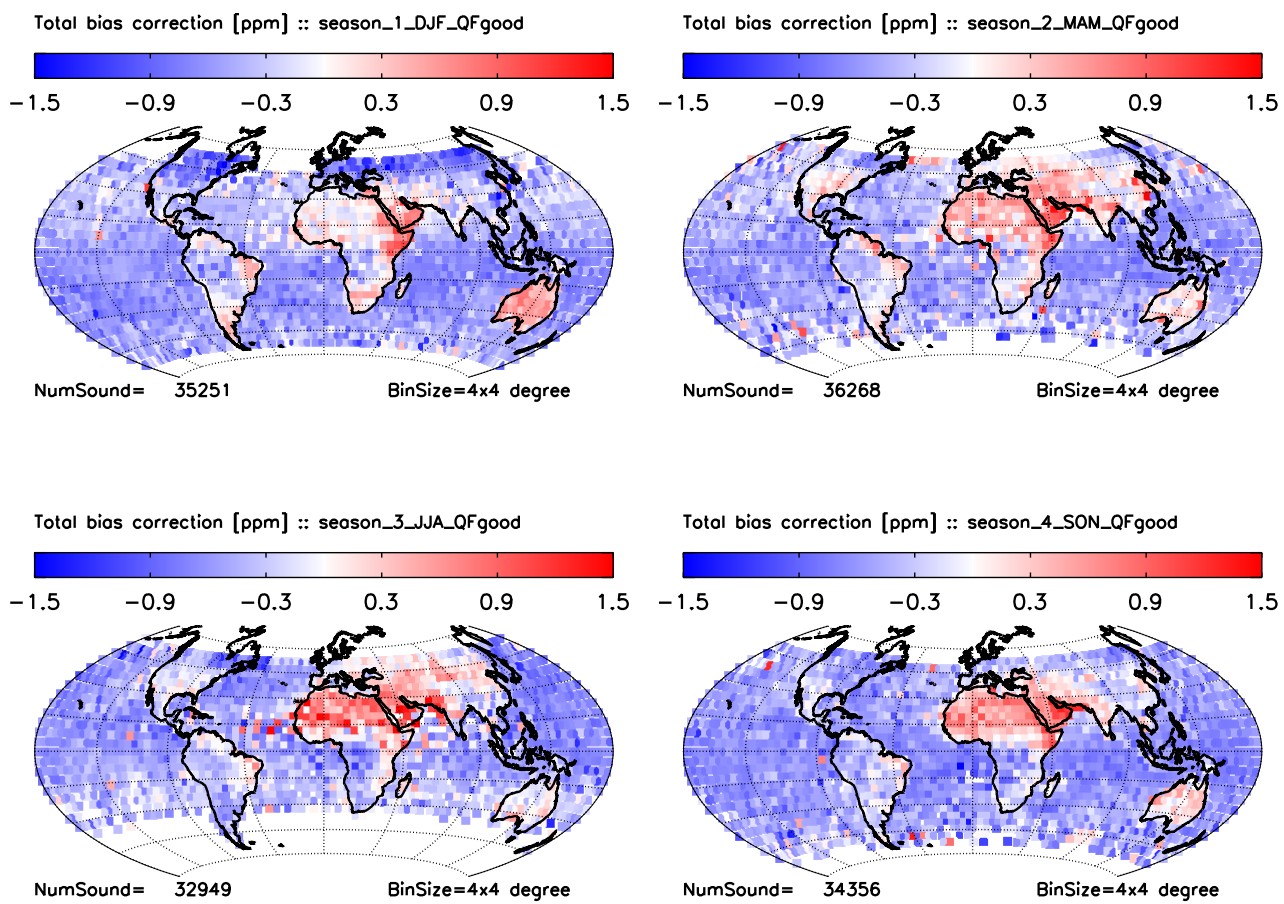

**Figure 15.** Seasonal maps of the total XCO$_2$ bias correction in 4° lat/lon bins for all good quality soundings. Top left is DJF, top right is MAM, bottom left is JJA and bottom right is SON.




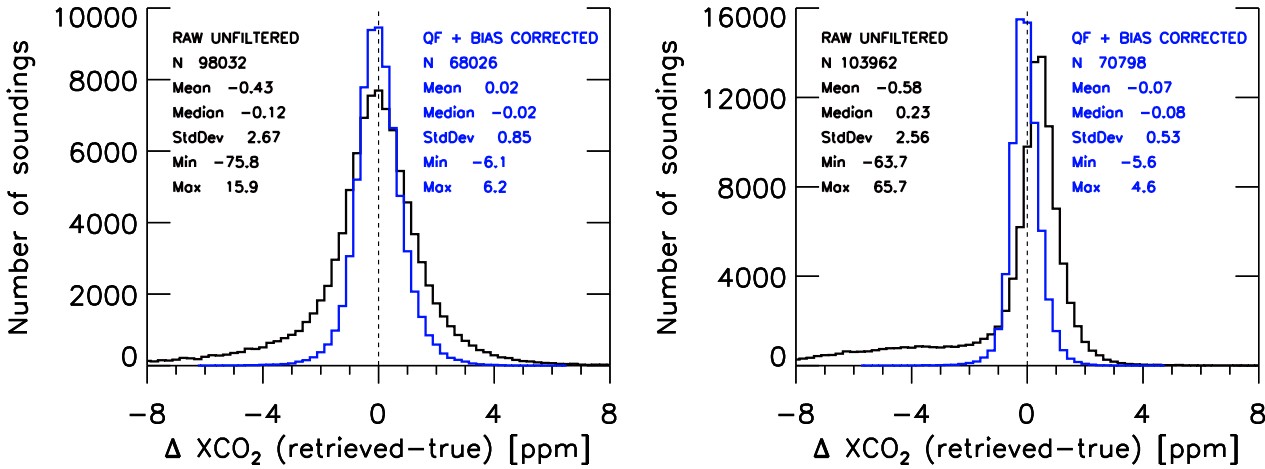

**Figure 16.** Histograms of the error in XCO$_2$ (retrieved - true) at 0.25 ppm resolution for land (left) and water (right) for the full year of simulations. The raw, uncorrected XCO$_2$ are shown in black, while the filtered and bias corrected values are shown in blue.




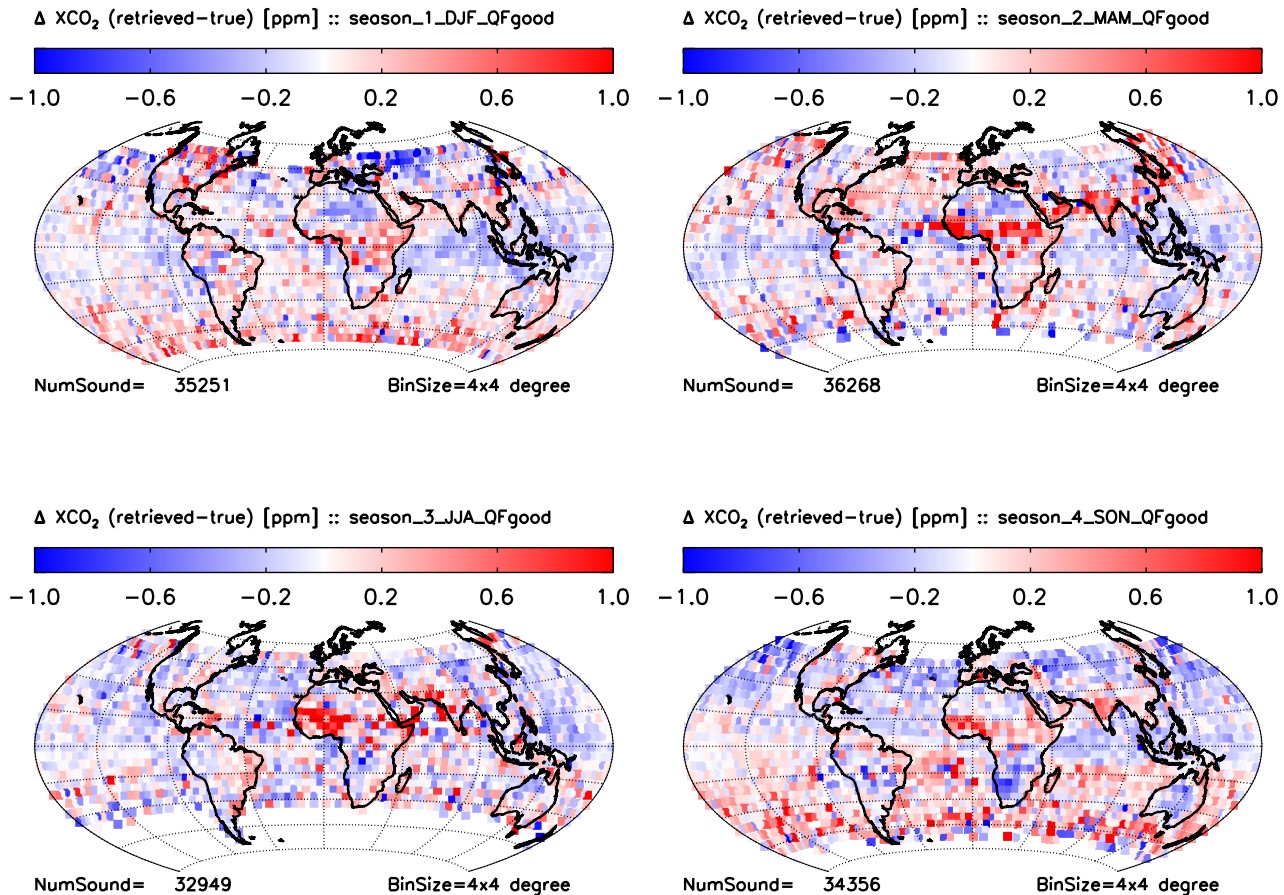

**Figure 17.** Seasonal maps of delta $XCO_2$ (retrieved - true [ppm]) in 4° lat/lon bins after quality filtering and bias correction. Top left is DJF, top right is MAM, bottom left is JJA and bottom right is SON.



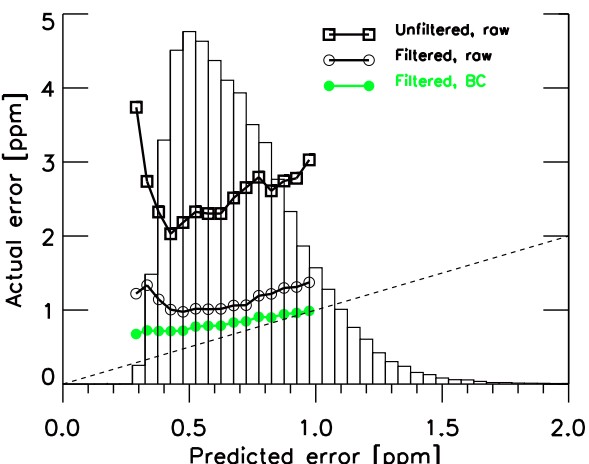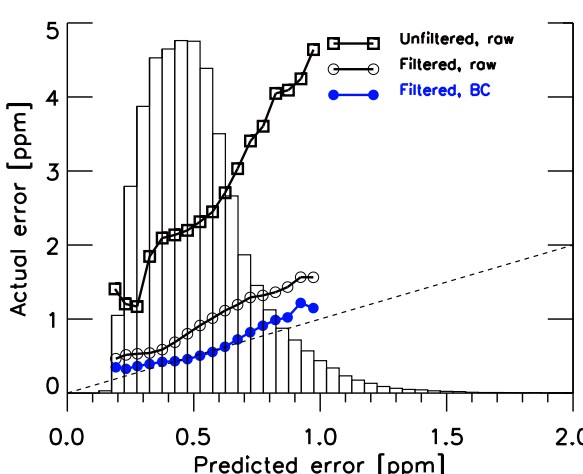

**Figure 18.** Binned median values of the actual versus predicted $XCO_2$ error for the full annual data set, for nadir-land (left panel) and the glint-water soundings (right panel). Each panel shows the unfiltered raw $XCO_2$ (open diamonds), the quality filtered raw $XCO_2$ (open circles), and the quality filtered and bias corrected $XCO_2$ (closed circles). To provide reference, the raw, unfiltered data set is also displayed as individual values (tiny black dots) and a histogram. The one-to-one line is shown.





**Figure 19.** Results comparing the IDP retrieved and true L1b SIF. These results are from the cloudy with noise-corrected SIF at 757 nm. The top two rows show maps gridded to $1°$ by $1°$ for the quality filtered JJA data set. The true L1b values (left) and the corresponding IDP retrieved values (right) are both in units W/m$^2$/$\mu$m/sr. The middle row shows the absolute difference between the retrieved and true values (left) and the fractional difference after an additional screening on true SIF $>0.2$ (right). The bottom row shows the linear correlation, along with some basic statistics for the full JJA set and the subsetted data.





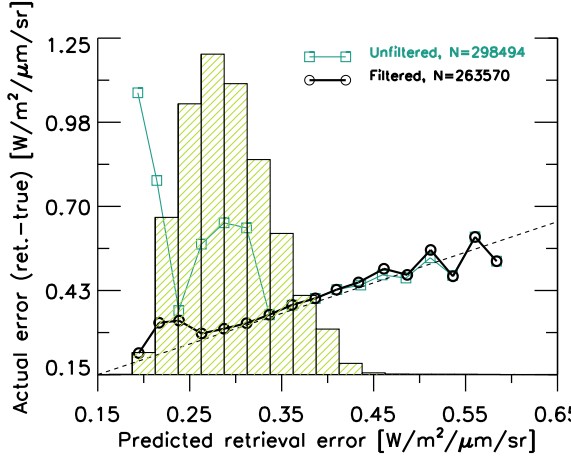

**Figure 20.** Actual versus predicted IDP SIF error for the unfiltered and filtered annual data set. This is for the nadir-land retrievals using instrument noise. The histogram shows the filtered data frequency normalized to one, with the one-to-one relationship given by the dotted line.

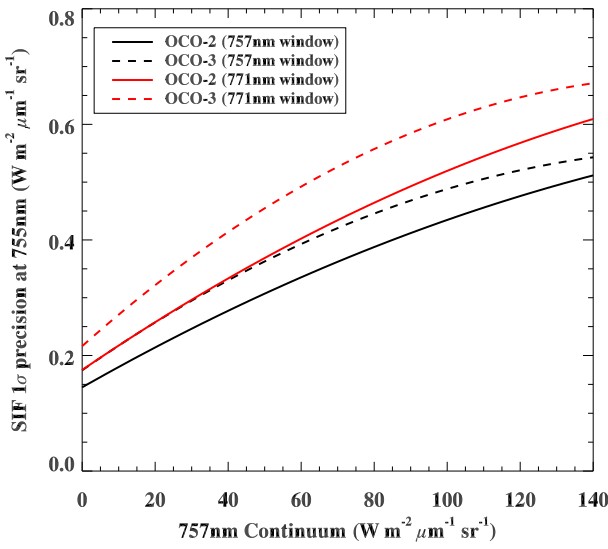

**Figure 21.** $1\sigma$ precision of estimated SIF, using both the 757 nm window (black) and the 771 nm window (red), for OCO-2 (solid) and OCO-3 (dashed). SIF precision at 755 nm was estimated from the 757 nm (771 nm) value by scaling by a factor of 1.10 (1.76), following Frankenberg et al. (2015). The OCO-2 values have been evaluated from the IDP posterior uncertainties in the actual SIF data product, for an average of all eight footprints. The OCO-3 values have come from an early version of the preflight noise estimates, which are subject to change.



**Table 1.** Summary statistics of the filtering for each stage of the analysis. Results are shown for the nadir-land, glint-water, and combined soundings separately.

| Filter | N Combined | N Land | N Water | Fraction passing (relative to combined total) Combined | Land | Water | Fraction passing (relative to surface type total) Land | Water |
|---|---|---|---|---|---|---|---|---|
| L1b (all) | 982922 | 337211 | 645711 | 100.0% | 34.3% | 65.7% | 100.0% | 100.0% |
| ABP (pass) | 237736 | 107295 | 130441 | 24.2% | 10.9% | 13.3% | 31.8% | 20.2% |
| L2 (converge) | 196211 | 96182 | 100029 | 20.0% | 9.8% | 10.2% | 28.5% | 15.5% |
| QF (good) | 140741 | 68264 | 72477 | 14.3% | 6.9% | 7.4% | 20.2% | 11.2% |



**Table 2.** Variables and thresholds used for quality filtering in the OCO-3 simulations for nadir-land. The cumulative fraction of scenes passing are also given. Note that the need for the water vapor scale factor (L2 WV scale) is likely due to a recently discovered bug in the simulator code that introduced a mismatch between the vertical profile in the scene and meteorology files.

| Variable Name | QF range | Cum Frac Pass |
|---|---|---|
| IDP $CO_2$ ratio | [0.9, 1.03] | 88.7% |
| IDP $H_2O$ ratio | [0.88, 1.1] | 85.6% |
| L2 dp | [-6.0, 7.0] | 81.6% |
| L2 total AOD | [0.0, 0.4] | 79.1% |
| L2 water AOD | [0.0008, 0.1] | 77.3% |
| L2 fine mode AOD | [0.0, 0.08] | 74.8% |
| L2 WV scale | [0.84, 0.95] | 73.7% |
| ABP dp | [-25.0, 4.0] | 71.7% |
| $CO_2$ grad del | [-40.0, 40.0] | 71.2% |
| L2 $\chi^2$ $O_2A$ | [0, 1.25] | 71.1% |
| L1b signal 31 | [0.075, 0.4] | 70.7% |





**Table 3.** Same as Table 2, but for glint-water soundings.

| Variable Name | QF range | Cum Frac Pass |
|---|---|---|
| Albedo slope weak $CO_2$ | [0.1, 100.0] | 84.3% |
| ABP dp | [-50.0, -3.0] | 81.7% |
| L2 dp | [-5.0, 2.0] | 75.0% |
| IDP $CO_2$ ratio | [1.0005, 1.015] | 73.5% |
| IDP $H_2O$ ratio | [0.88, 1.03] | 72.6% |
| L2 $CO_2$ grad del | [-30.0, 60.0] | 72.4% |
| L2 Total AOD | [0.0, 0.25] | 72.2% |
| Solar zenith angle | [0.0, 63.0] | 70.4% |



**Table 4.** Bias correction parameters for the the OCO-3 simulation. Only the dp terms have units [hPa], while the other parameters are unitless. Again, the need for a water vapor scaling factor bias correction term is likely due to an indexing bug in the L1b simulator code. The "DS" AOD is the combined optical depth of dust and sea salt aerosols.

| Variable | Nadir-land<br>Global bias=0.18<br>BC slope, offset | Glint-water<br>Global bias=0.0<br>BC slope, offset |
|---|---|---|
| L2 dp | -0.20, 1.0 [hPa] | -0.21, 0.0 [hPa] |
| L2 WV scale | 14.0, 0.9 | NA |
| L2 DS AOD | -7.6, 0.0 | NA |
| L2 Fine mode AOD | 14.0, 0.0 | NA |
| IDP $CO_2$ ratio | NA | -170.0, 1.003 |
| ABP dp | NA | -0.053, 0.0 [hPa] |





**Table 5.** The slope and offset of a linear least squares fit (LLS) and Pearson linear correlation coefficient (R) for $\Delta XCO_2$ versus geophysical parameters before and after application of the QF/BC procedure. Data is for the full annual glint-water soundings only.

| Variable Name | Pre QF/BC | | Post QF/BC | |
|---|---|---|---|---|
| | LLS m/b | R | LLS m/b | R |
| Sounding latitude | -0.000/ 0.520 | -0.009 | 0.002/-0.086 | 0.041 |
| Solar zenith angle | -0.013/0.924 | -0.255 | -0.002/0.009 | -0.052 |
| Polarization angle | -0.016/ 0.928 | -0.256 | -0.002/ 0.009 | -0.052 |
| SNR (Oxygen-A band) | -0.000/ 0.513 | -0.001 | -0.000/-0.054 | -0.002 |
| SNR (Weak $CO_2$ band) | -0.000/ 0.519 | -0.005 | -0.000/-0.054 | -0.001 |
| SNR (Strong $CO_2$ band) | -0.000/ 0.518 | -0.005 | -0.000/-0.055 | -0.001 |
| True AOD | -1.469/ 0.673 | -0.171 | -0.029/-0.053 | -0.004 |



**Table 6.** Comparison of the standard deviation in $XCO_2$ before and after QF and BC.

|               | Land-nadir | Ocean-glint |
| ------------- | ---------- | ----------- |
| N             | 96,182     | 100,029     |
| Sigma raw     | 1.88 ppm   | 2.15 ppm    |
| Sigma BC      | 1.79 ppm   | 1.76 ppm    |
| Sigma raw QF  | 1.14 ppm   | 0.67 ppm    |
| Sigma QF, BC  | 0.85 ppm   | 0.52 ppm    |

## 6  Summary

The work presented here highlights the overall science objectives and expected performance for NASA's upcoming Orbiting Carbon Observatory-3 (OCO-3) mission. OCO-3 will be a hosted payload on the International Space Station, which is in a precessing orbit. The launch is currently planned for early 2019, with a nominal three year mission life. While the instrument itself is a duplicate of the operational OCO-2, several features, such as the addition of a pointing mirror assembly, and other necessary optical components, will slightly alter the instrument performance. The OCO-3 mission will largely inherit the data algorithms that have been tested and refined using OCO-2.

After introducing the high level science objectives (which are similar to the OCO-2 mission) and providing a brief overview of the planned measurement strategy, a detailed analysis of a year long simulation of OCO-3 measurements is presented. The analysis begins with realistic ephemeris and measurement geometries, which are used, along with modeled meteorology and trace gases, to generate synthetic L1b radiances. Cloud screening preprocessors are used to select soundings to be run through the OCO-2 B8 L2 full physics retrieval, which is the current version of the algorithm that will be adopted for the OCO-3 mission. We performed a full quality filtering and bias correction to the retrieved L2 $XCO_2$, following the methodology given by O'Dell et al. (2018). Overall $XCO_2$ errors, relative to both the true $XCO_2$ and to the predicted error, are assessed. We also present analysis of the solar induced chlorophyll fluorescence from the IMAP DOAS retrieval algorithm, and discuss implications of the spatiotemporal sampling from the ISS relative to polar orbiters.

Generally, these simulations highlight the spatial and temporal sampling expected from OCO-3 aboard the precessing ISS, illustrating how measurements will span a wide range of sunlit hours, and have large day to day variation in latitudinal sampling. The simulated L1b radiances show signal to noise characteristic that are generally slightly lower than OCO-2, but still sufficient to accurately estimate $XCO_2$. Over monthly timescales that are typical of the global flux analysis, all latitudes are sampled and roughly 2.5 million good quality L2 retrievals are expected. An assessment of the error characteristics of $XCO_2$ indicate that they will be comparable to operational OCO-2 data. Furthermore, we demonstrate that the general methodology of L2 quality filtering and bias correction on the retrieved $XCO_2$ which is being used on operational OCO-2 data can be used to identify the



most useful data and reduce the bias inherent in the full physics retrieval algorithm. In fact, the filtering and bias correction process is necessary in order to meet the measurement objectives.

Retrievals of SIF using the IDP algorithm are also expected to have similar error characteristics compared to OCO-2, especially with respect to error induced by instrument noise. This new set of space-based SIF will be highly informative because of the varying time of day sampling, so important to characterizing the local behavior of SIF. The dense coverage of latitudes below about $50°$, where global SIF is most active on an annual scale, is expected to provide a rich data set to the science community.

Overall, the OCO-3 performance characteristics, as assessed in this simulation, should provide a global dataset that achieves the mission goals and continues the dense, high-precision $XCO_2$ and SIF record from space-borne measurements.



*Author contributions.* The overall research goals were crafted by AE, with contributions from CWO and TET. RP provided the OCO-3 simulated geometry, with input from AE. TET generated the L1b radiances and ran the ABP and IDP preprocessors. AE ran the L2 FP retrievals and CWO performed the quality filtering and bias correction procedure. The data analysis and word smithing were joint efforts between TET, CWO and AE. The figures were generated by TET, with input from CWO and AE.

5 *Competing interests.* We declare no competing interests in this work.

*Acknowledgements.* A portion of the research described in this paper was carried out at the Jet Propulsion Laboratory, California Institute of Technology, under a contract with the National Aeronautics and Space Administration. The CSU contribution to this work was supported by JPL subcontract 1439002.

We would like to acknowledge the hard work of the OCO-3 calibration team for providing data necessary to generate the L1b simulations.



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
