# Peer review of "The OCO-3 mission; measurement objectives and expected performance based on one year of simulated data"

_Atmospheric Measurement Techniques, 2018_

## Referee Comment (RC1) · Anonymous Referee #1 · 16 Dec 2018

This manuscript presents an analysis of the capabilities and performance of the OCO-3 CO2 satellite instrument, a carbon copy of OCO-2, scheduled to be deployed on the International Space Station (ISS) early in 2019. Based on one complete year of simulated L1B data, the manuscript describes in detail the expected coverage in time and space (very different from OCO-2 due to the precessing orbit of the ISS), signal:noise ratios of the nadir (over land) and glint (over oceans) observations, the effects of the pointing mirror assembly (PMA) not present on OCO-2, and the performance of L2 data (column average CO2) derived from the synthetic measurements using a full physics retrieval. Some analysis of the solar induced fluorescence data is also presented. The manuscript is relevant and timely and will be an important reference for future OCO-3

publications. Writing and content are of high quality, although the paper is somewhat lengthy and not always very exciting to read due to its technical and documentary nature.

I clearly support publication after addressing my (embarrassingly small number of) minor points:

Section 3.2 presents the issue of polarization and the computation of Stokes coefficients, but no further analysis is presented in the results section. Why should the reader be interested in the computation of these Stokes coefficients (Eq. 1 and 2) if they are not used later on? It would be nice to actually see some discussion of polarization effects in Section 5.1, as promised at the end of Sect. 3.2.

Page 4, Line 25: Isn't the O2-A band also providing useful information on aerosols?

P15, L19: I didn't quite understand how exactly the random selection of cloud and aerosol profiles was made. Was the random selection made following some probability of detecting a cloud based on climatological cloud coverage?

Small corrections:

1. Page 3, line 32: 'then 4 km' -> 'than 4 km'

2. P3, L12. Point is missing in front of 'Finally'

3. P9, L14: It should probably be '30° south latitude'

4. P9, L26: It should probably be '2° longitude x 2° latitude'. Same issue in legend of Fig. 3.

5. P14, L12: There are two 'a' on this line that should be deleted.

6. P16, L15: I suggest to delete 'accurately' in this sentence. The calculation will always only be an approximation of reality and will only be (nearly) accurate, if the inputs such as aerosol properties are accurate.

7. P17, L1: What is the 'NASA DISC'?

8. P17, L13: What do you mean by ' a priori state of the atmosphere'? What properties of the atmosphere are described?

9. P18, L29: 'anomoly' -> 'anomaly'

10. P19, L32: delete one of the two 'when'

11. P20, L6: There seems to be a closing bracket missing.

12. P20, L7: 'the OCO-3 SNR' -> 'the SNR'

13. Tables 2-4: These tables are full of acronyms and hard to read. I suggest adding another column "description" describing the variables.

14. P25, L2: Could you be more specific regarding the similarities and differences in the selected variables between OCO-3 and OCO-2¿

15. Fig. 13, lower left panel: Why is the x-axis the square root of the AOD and not AOD directly?

16. Fig. 18: I can't see any 'tiny black dots'

---

## Referee Comment (RC2) · Anonymous Referee #2 · 8 Feb 2019

This paper provides a detailed assessment of the expected performance of the Orbiting Carbon Observatory-3 (OCO-3) mission, in which a direct copy of the OCO-2 shortwave infrared (SWIR) grating spectrometer will be installed on the International Space Station (ISS). Whilst the instrument sensitivity and performance characteristics are therefore expected to be very similar to those of OCO-2 (as referenced in the Introduction section), there are a couple of important differences owing to use of the ISS as a platform for the instrument. Firstly the ISS has a precessing orbit, resulting in the 'significant differences from day to day in the sampling locations and time of day'. The analysis of the spatial and temporal sampling expected from the ISS orbit forms the bulk of this paper. Secondly, the OCO-3 instrument will have a new pointing mirror as-

sembly (PMA) allowing rapid switching between nadir and glint observation modes, as well as target and snapshot modes for dwelling over specific locations on the ground, without having to re-orient the spacecraft. The paper focuses on the two primary viewing modes – nadir-land and glint-water – and correctly leaves the snapshot mode to a companion paper, given the level of detail already present here.

The findings of the paper are based on a year-long simulated OCO-3 dataset, which uses atmosphere and surface model output along with an instrument model based on laboratory testing. The simulated observation geometry uses actual ISS ephemeris data from the year 2015 to take into account the spacecraft position and velocity at a rate of 1Hz. It is not stated whether the spacecraft pitch and roll is either taken into account (i.e. the PMA can compensate for spacecraft pitch and roll in real time) or is assumed to be small enough that the results aren't affected; it would be interesting to know whether the spacecraft orientation is a problem in terms of achieving the required pointing accuracy. The spatiotemporal distribution of the simulated soundings is clearly illustrated in Figures 2 through 5, neatly showing the advantages and limitations of this sampling pattern. Figures 4 and 5 are particularly interesting since they show how the sounding time-of-day varies over the year, in contrast to the polar orbiting sun synchronous OCO-2 and GOSAT missions. The full physics retrieval algorithms used to obtain XCO2 and solar induced fluorescence (SIF) are well established (as can be seen through the references given in Section 4), and the same as those used in OCO-2 operational data processing.

Section 5 provides a useful comparison between the expected OCO-3 performance and that of OCO-2, at both the spectral radiance (Level 1B) and retrieved XCO2 (Level 2) stages. Whilst OCO-3 is expected to have a worse signal-to-noise ratio performance in the CO2 bands (particularly the 1.6 micron band) than OCO-2, owing to a higher detector noise floor and decreased signal incurred by different fore-optics including the PMA, the expected SNR values are still sufficiently high to achieve the desired XCO2 retrieval precision. In the spatial comparison shown in Figure 8, the lack of OCO-2

soundings over the Amazon, sub-Saharan Africa, and China compared with OCO-3 stands out; is this due to the 'deficiencies in the simulation setup' mentioned on page 19 line 28, meaning that in reality we might expect these regions to be filtered out in the OCO-3 data as well as in OCO-2?

The results section concludes with a thorough assessment of the error characteristics of the simulated OCO-3 dataset. The SIF dataset looks as though it will be particularly interesting, the increased coverage at lower latitudes and soundings at different times of day compensating for the worse precision than that achievable with OCO-2. Overall, this work provides a valuable resource for those intending to work with the OCO-3 dataset, and I suggest that it is ready for publication once the questions raised above are addressed, along with the following minor points:

Page 2 Line 7: GOSAT-2 has now been launched, on 29th October 2018

Page 6 Line 4: Update on whether analysis of thermal vacuum testing data is complete (I assume the results will go in the forthcoming manuscript mentioned on Line 5)

Page 18 Line 23: I assume this refers to the same testing data mentioned on Page 6 Line 4; perhaps an 'in preparation' reference would help clarify this?

Page 20 Line 6: Close brackets on 'Figure 7'

Page 25 Line 18: Acronym 'IDP' not defined?

Page 25 Line 25: Move 'e.g.' inside the brackets, i.e. '... gases from space (e.g. Aben et al., 2007; Butz et al., 2009)'

---

## Author Comment (AC1) · 5 Mar 2019

**The OCO-3 mission; measurement objectives and expected performance based on one year of simulated data**
**Response to Anonymous Reviewer 1**

Annmarie Eldering[1], Tommy E. Taylor[2], Chris W. O'Dell[2], and Ryan Pavlick[1]

[1]Jet Propulsion Laboratory, California Institute of Technology, Pasadena, CA, 91109, USA.
[2]Cooperative Institute for Research in the Atmosphere, Colorado State University, Fort Collins, CO, 80521, USA.

Thank you for the very useful comments on our work. We appreciate your time and interest. The reviewer's comments are in black text and our responses are given in blue.

We'd like to note that we took the opportunity to make a few minor changes and improvements to the manuscript during the response-to-reviewers process. Namely the discussion in various places was abbreviated in an attempt to improve the overall flow of the manuscript. The most significant edit was made in Section 3.1 "Simulated OCO-3 observation geometry". Also, a few minor text changes were made throughout to reflect slight updates to the knowledge of expected OCO-3 operations since the time of the original submission in early October, 2018 (nearly 5 months ago).

1. Section 3.2 presents the issue of polarization and the computation of Stokes coefficients, but no further analysis is presented in the results section. Why should the reader be interested in the computation of these Stokes coefficients (Eq. 1 and 2) if they are not used later on? It would be nice to actually see some discussion of polarization effects in Section 5.1, as promised at the end of Sect. 3.2.

   This is a fair point. We felt that some discussion of the polarization angle is warranted as this to some extent drives the "throughput" or "signal" that the instrument measures. Since the polarization angle is a form of geometry, we felt that placement in Section 3 "Simulated geometry, meteorology and L1b dataset" was the appropriate place to introduce it. In an attempt to better tie the discussion of the (expected) OCO-3 polarization angle to the solar zenith angle and the signal to noise ratio, we have provided a more comprehensive discussion of the polarization in Section 3.2 "Simulated instrument polarization angle and Stokes coefficients". This includes addition of a new figure (Fig.6) that shows the theoretical relationship between the polarization angle, the solar zenith angle and the measured signal to noise ratio for a specularly reflecting surface driven by a Cox and Munk model, i.e., glint-water viewing. The reference in Sec 3.2 to further discussion on polarization in Sect 5.1 was removed since Sect 5.1 focuses mainly on the SNR aspect of the measurements, which is of overall greater importance than polarization w/r/t $XCO_2$ retrievals.

2. Page 4, Line 25: Isn't the O2-A band also providing useful information on aerosols?

   Yes, this is true. We have slightly modified the wording in Section 2.1: "The OCO-3 instrument payload" to more accurately describe the function of each of the three spectral bands. A few relevant citations have been added as well.

[Figure]

**Figure 1.** Maps showing the number of CALIPSO cloud and aerosol profiles contained in each $2°$ by $2°$ lat/lon bin for January (left) and September (right). For each OCO-3 sounding in the simulated data set a single profile was picked at random from the appropriate month and bin.

3. P15, L19: I didn't quite understand how exactly the random selection of cloud and aerosol profiles was made. Was the random selection made following some probability of detecting a cloud based on climatological cloud coverage?

No, there is no climatological cloud coverage taken into account. However, the static database is composed of real CALIPSO/CALIOP profiles that are binned into monthly 2x2 lat/lon bins, so each randomly selected profile represents a scene that was actually measured by CALIOP, but there is no continuity among adjacent OCO-3 soundings which are order 2x2 km. Some examples of the number of CALIPSO profiles for each lat/lon bin are shown in Figure 1. Although this material is interesting, we feel that the paper is already a bit lengthy so have opted not to include this figure in the text. Some alterations were made to the wording in Section 3.3 "Simulated meteorology, gas and cloud/aerosol fields" in an attempt to more clearly explain the procedure.

4. Small corrections:

    (a) 1. Page 3, line 32: 'then 4 km' -> 'than 4 km' Corrected.

    (b) 2. P3, L12. Point is missing in front of 'Finally' Corrected.

    (c) 3. P9, L14: It should probably be '30 south latitude' Corrected.

    (d) 4. P9, L26: It should probably be '2 longitude x 2 latitude'. Same issue in legend of Fig. 3. Corrected.

    (e) 5. P14, L12: There are two 'a' on this line that should be deleted. Corrected.

    (f) 6. P16, L15: I suggest to delete 'accurately' in this sentence. The calculation will always only be an approximation of reality and will only be (nearly) accurate, if the inputs such as aerosol properties are accurate. Corrected.

    (g) 7. P17, L1: What is the 'NASA DISC'? Removed this reference as it is irrelevant in this context.

    (h) 8. P17, L13: What do you mean by ' a priori state of the atmosphere'? What properties of the atmosphere are described? Slight rewording of Section 4.2 to better describe the setup of the L2 retrieval algorithm.

(i) 9. P18, L29: 'anomoly' -> 'anomaly' Corrected.

(j) 10. P19, L32: delete one of the two 'when' Corrected.

(k) 11. P20, L6: There seems to be a closing bracket missing. Corrected.

(l) 12. P20, L7: 'the OCO-3 SNR' -> 'the SNR' Corrected.

(m) 13. Tables 2-4: These tables are full of acronyms and hard to read. I suggest adding another column 'description' describing the variables. Updated the three tables to include a Variable Description column

(n) 14. P25, L2: Could you be more specific regarding the similarities and differences in the selected variables between OCO-3 and OCO-2 We modified the discussion in Section 5.4 "Bias correction of $XCO_2$" to be more explicit about the similarities and differences with both real and simulated OCO-2 data.

(o) 15. Fig. 13, lower left panel: Why is the x-axis the square root of the AOD and not AOD directly? We found that the $XCO_2$ error was closer to linear versus the square-root of the AOD. Since the BC formulation is linear in nature, this is the optimal variable to make the correction against. Table 4 was corrected to show this variable (was improperly just DS AOD before). It is such a minor detail that we did not insert any additional verbiage into the text for the sake of brevity.

(p) 16. Fig. 18: I can't see any 'tiny black dots' Removed the reference to the tiny black dots since they had been removed from this version of the plot for clarity.

---

## Author Comment (AC2) · 5 Mar 2019

**The OCO-3 mission; measurement objectives and expected performance based on one year of simulated data**
**Response to Anonymous Reviewer 2**

Annmarie Eldering[1], Tommy E. Taylor[2], Chris W. O'Dell[2], and Ryan Pavlick[1]

[1]Jet Propulsion Laboratory, California Institute of Technology, Pasadena, CA, 91109, USA.
[2]Cooperative Institute for Research in the Atmosphere, Colorado State University, Fort Collins, CO, 80521, USA.

Thank you for the very useful comments on our work. We appreciate your time and interest. The reviewer's comments are in black text and our responses are given in blue.

We'd like to note that we took the opportunity to make a few minor changes and improvements to the manuscript during the response-to-reviewers process. Namely the discussion in various places was abbreviated in an attempt to improve the overall flow of the manuscript. The most significant edit was made in Section 3.1 "Simulated OCO-3 observation geometry". Also, a few minor text changes were made throughout to reflect slight updates to the knowledge of expected OCO-3 operations since the time of the original submission in early October, 2018 (nearly 5 months ago).

1. It is not stated whether the spacecraft pitch and roll is either taken into account (i.e. the PMA can compensate for spacecraft pitch and roll in real time) or is assumed to be small enough that the results aren't affected; it would be interesting to know whether the spacecraft orientation is a problem in terms of achieving the required pointing accuracy. This is a good point. Yes, the PMA does compensate for ISS pitch and roll. We updated the text in Section : "Sampling from the International Space Station - routine measurements" to better describe the expected operations and provided a reference to an ISS technical document that specifies orbit parameters and other interesting information.

2. In the spatial comparison shown in Figure 8, the lack of OCO-2 soundings over the Amazon, sub-Saharan Africa, and China compared with OCO-3 stands out; is this due to the 'deficiencies in the simulation setup' mentioned on page 19 line 28, meaning that in reality we might expect these regions to be filtered out in the OCO-3 data as well as in OCO-2? Yes, our expectation is that the OCO-3 data as presented here is a bit over optimistic relative to what is expected from real data. Think of it as a "best case" scenario, if all systems function perfectly, e.g., perfect instrument calibration and spectroscopy. Some text has been added to the discussion in Section 5.1 "Simulated L1b radiance characteristics" in reference to Fig 8 to more explicitly call out and explain the expected lack of data in these three particular regions.

3. Page 2 Line 7: GOSAT-2 has now been launched, on 29th October 2018 We slightly modified the discussion in the Introduction to more accurately describe the current situation as the original text was a bit out of date.

4. Page 6 Line 4: Update on whether analysis of thermal vacuum testing data is complete (I assume the results will go in the forthcoming manuscript mentioned on Line 5) The final OCO-3 TVAC data is still under analysis and the results are

planned for publication, likely post-launch. The discussion in Section 2.2 "OCO-3 pointing mirror assembly overview" was updated and consolidated in Section 5.1 "Simulated L1b radiance characteristics" to more accurately describe the current situation as the original text was a bit out of date.

5.  Page 18 Line 23: I assume this refers to the same testing data mentioned on Page 6 Line 4; perhaps an 'in preparation' reference would help clarify this? To clarify the message, the discussion of instrument calibration was removed from Section 2.2 "OCO-3 pointing mirror assembly overview" and consolidated in Section 5.1 "Simulated L1b radiance characteristics".

6.  Page 20 Line 6: Close brackets on 'Figure 7' Corrected.

7.  Page 25 Line 18: Acronym 'IDP' not defined? The IMAP-DOAS Preprocessor (IDP) was first defined in Section 4.1 "Preprocessors". We now spell it out explicitly again in the header of Section 5 for the convenience of the reader.

8.  Page 25 Line 25: Move 'e.g.' inside the brackets, i.e. '. . . gases from space (e.g. Aben et al., 2007; Butz et al., 2009)' Corrected.